# L2RBOX: LINE-SUPERVISED ORIENTED OBJECT DETECTION

## ABSTRACT

Oriented object detection is crucial for complex scenes such as aerial images and industrial inspection, providing precise delineation by minimizing background interference. Recently, the weakly-supervised oriented object detection has gaining attention due to its cost-effectiveness. However, the majority of existing weakly-supervised methods are either point-supervised or HBox-supervised, which presents a challenge in achieving an optimal balance between annotation cost and detection performance. In response, we introduce a novel form of line annotation, which is intermediate between point-level and plane-level annotation. Based on this, we present L2RBox, an end-to-end anchor-free detector that is the first line-supervised method for oriented object detection. The fundamental objective of the L2RBox is to utilise line labels for the completion of label assignment and the calculation of loss. In particular, the line is mapped to the corresponding circle domain, which is then used to select training samples and calculate the center-ness target by the minimum circumscribed rectangle of the circle in the direction of the line. The regression loss that we propose is designed to support the line as an optimisation target. It comprises four components, namely scale loss $L_s$, height loss $L_h$, position loss $L_p$ and angle loss $L_a$. Extensive experimentation on DOTA-v1.0 and DIOR-R has demonstrated that our L2RBox significantly outperforms point-supervised methods, while requiring only a slight increase in labeling costs. It is also noteworthy that the proposed approach also demonstrates a slight performance advantage over the fully-supervised FCOS in certain categories.

## 1 INTRODUCTION

In recent years, oriented object detection has progressed rapidly, leveraging advancements in horizontal object detection Liu et al. (2020). Its fine-grained rotated bounding box (RBox) has proven highly effective in complex scenarios such as aerial imagery, scene text, and industrial inspection Wen et al. (2023). Although detectors have made significant progress with extensive annotated data, full supervision in oriented object detection faces several challenges: the RBox annotation format is less prevalent in many existing datasets, and producing RBox annotations is more expensive.

To mitigate the dependence on labor-intensive RBox labeling, weakly-supervised object detection represents a solution. As illustrated in Fig. 1, existing weakly-supervised methods employ coarser-grained annotations as weakly-supervised signals to predict RBox, which are roughly divided into point-supervised methods and HBox-supervised methods according to the annotation. For HBox-supervised methods, H2RBox Yang et al. (2023) and H2RBox-v2 Yu et al. (2024b) have explored the HBox-to-RBox setting that learns RBox detectors from horizontal bounding box (HBox) annotation. However, Plane-level annotations are still inefficient and labor-intensive. Therefore, point-supervised methods PointOBB Luo et al. (2024) and Point2RBox Yu et al. (2024a) have further explored more cost-effective point-level annotation forms, but these methods suffer from lower detection accuracy and often require additional knowledge or pseudo-label generation. In summary, these weakly-supervised methods have complex structures and cannot balance detection accuracy and annotation costs.

In response, we trade off the annotation cost with the detection accuracy for weakly-supervised methods and first propose line annotation format for oriented object detection. Specifically, we label the object along its central axis, and the process is flexible, allowing for some margin of error in the

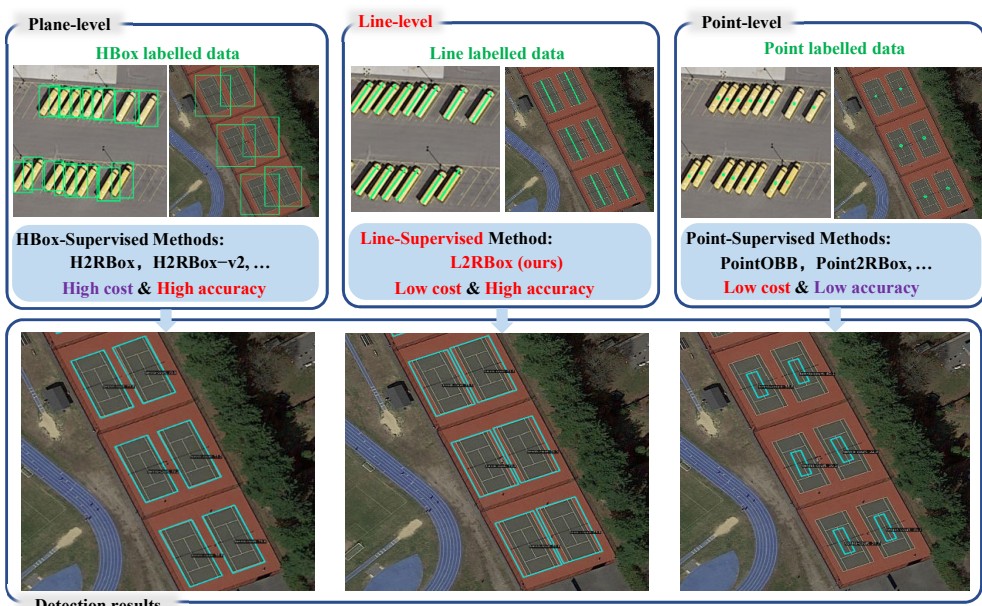

Figure 1: The top row visualizes examples of three weakly-supervised annotation forms, and the bottom row visualizes the detection results of the same scene. Our proposed line-supervised L2RBox (line-level) achieves both low cost and high accuracy.

line annotations. As shown in Fig. 1 top row, compared to the point, the line offers significantly richer information with only a slight increase in labeling costs. For a fair comparison, we use the annotation website[1] to evaluate different annotation formats, the average time for annotating 100 instances is 99.15s for point annotations, 178.8 s for line annotations, 332.7s for HBox, and 516.2s for RBox. Based on line annotations, the detection performance of our proposed L2RBox significantly outperforms Point-supervised methods, and is comparable to HBox-supervised methods, as shown in Fig. 1 bottom row. Additionally, to reflect the effectiveness in balancing annotation cost and performance of weakly-supervised methods, we designed a trade-off metric $M$ that considers both the accuracy and cost, emphasizing the balance between them. Section 4.2 provides a comprehensive analysis of the performance of weakly-supervised methods in terms of trade-off metrics.

The line-level annotation introduces a novel task setting: using line annotations to achieve significantly better performance than point-supervised methods. In this paper, we propose a simple yet effective approach dubbed as L2RBox, the first line-supervised oriented object detector. As an end-to-end anchor-free detector, the core of our L2RBox is to use line labels to complete label assignment and loss calculation, where label assignment includes training sample selection and center-ness target calculation. Specifically, we map the line to the corresponding circle domain and use this to select training samples and calculate the center-ness target by the minimum circumscribed rectangle of the circle in the direction of the line (see Fig. 2 top L-LA). In the regression branch (see Fig. 2 bottom Branches), our proposed regression loss supports the line as an optimization target comprising four components: scale loss $L_s$, height loss $L_h$, position loss $L_p$ and angle loss $L_a$. Extensive experiments demonstrate that our L2RBox achieves significantly better performance than point-supervised methods with only a slight increase in labeling costs. Meanwhile, our L2RBox achieves optimal results on the trade-off metric, indicating that our method can effectively balance detection accuracy and cost. Our main contributions are as follows:

**1)** To our best knowledge, we first propose line annotation format for oriented object detection, which trade-off the annotation cost with the detection accuracy in weakly-supervised methods.

**2)** We propose specialized end-to-end detectors for line supervision, including label assignment and loss functions that support line annotation, where label assignment includes training sample selection and center-ness target calculation.

---

[1] https://www.makesense.ai/

**3)** Extensive experiments on DOTA-v1.0 and DIOR-R show that our L2RBox far outperforms Point-supervised methods with only a slight increase in labeling costs *e.g.* our L2RBox achieves AP$_{50}$ of 58.26% on DOTA, which is an improvement of 28.18% over the point-level method PointOBB. Notably, it also offers a slight performance advantage over fully-supervised FCOS in some categories.

## 2 RELATED WORK

### 2.1 FULLY-SUPERVISED ORIENTED OBJECT DETECTION

Oriented object detection algorithms primarily focus on aerial objects, multi-oriented scene texts, retail, etc. Notable approaches in this field include the anchor-based detector Rotated RetinaNet Lin et al. (2017c), the anchor-free detector Rotated FCOS Tian et al. (2019), and two-stage detectors such as Oriented R-CNNXie et al. (2021), RoI Transformer Ding et al. (2019), and ReDet Han et al. (2021). To address the boundary problem caused by the periodicity of angles, RSDet Qian et al. (2021) proposes a modulation loss to alleviate loss jumps. CSL Yang & Yan (2020) and DCL Yang et al. (2021a) convert the angle into boundary-free coded data. GWD Yang et al. (2021b), KLD Yang et al. (2021c), and KFIoU Yang et al. (2022) propose Gaussian-based losses that convert RBox into a Gaussian distribution. PSC Yu & Da (2023) proposes a Phase-Shifting Coder that encodes the orientation angle into periodic phases. Additionally, RepPoint-based approaches Yang et al. (2019); Hou et al. (2023); Li et al. (2022a) provide new alternatives for oriented object detection by predicting a set of sample points that bounds the spatial extent of an object. In this study, in order to reduce the reliance on labor-intensive RBox labeling, we concentrate on the more challenging task of weakly-supervised oriented object detection.

### 2.2 WEAKLY-SUPERVISED ORIENTED OBJECT DETECTION

Existing mainstream weakly-supervised oriented object detection approaches can be divided into HBox-supervised (plane-level) and point-supervised (point-level) methods. Furthermore, we explore the feasibility of line-supervised (line-level) methods.

**HBox-supervised.** HBox-supervised instance segmentation methods Tian et al. (2021); Li et al. (2022b); Kirillov et al. (2023) employ the HBox-Mask-RBox pipeline to derive RBox from the segmentation mask, though this is less cost-effective. A pioneering approach, H2RBox Yang et al. (2023), bypasses the segmentation step and directly detects RBox from HBox annotations. As a new version, H2RBox-v2 Yu et al. (2024b) exploits the inherent symmetry of objects. EIE Wang et al. (2024) leverages various contrastive cues related to angle prediction, facilitating the learning of equivariance between boxes. Nevertheless, these techniques still necessitate the acquisition of a considerable number of bounding box annotations. Additionally, OAOD Iqbal et al. (2021) uses extra object angle, whereas KCR Zhu et al. (2023) employs RBox-annotated source datasets with HBox-annotated target datasets. However, these specialized annotation forms lack universality.

**Point-supervised.** Point-based annotations have been widely used in horizontal object detection Chen et al. (2021); Ying et al. (2023). Due to its cost-effectiveness and efficiency, point-supervised oriented object detection has garnered attention. P2BNet Chen et al. (2022) uses Multiple Instance Learning (MIL) to select the box with the highest confidence from multiple boxes containing points. Point-to-Mask Li et al. (2023) provides a potential method for point-supervised rotating target detection. PointOBB Luo et al. (2024) learns object scale and angle information through self-supervised learning across different views, enabling the generation of oriented bounding boxes from points. Point2RBox Yu et al. (2024a) transfers object features to synthetic patterns using a sampling strategy and trains output RBox on transformed images, enhancing the network's perception of size and rotation for improved detection accuracy. However, these methods suffer from lower detection accuracy and often require additional knowledge or pseudo-label generation.

**Line-supervised.** In light of the aforementioned methods, we have sought to strike a balance between the annotation cost and the detection accuracy in weakly supervised methods. To this end, we explore the potential of utilizing line as a means of labeling objects, which has a cost between point-level and plane-level. This paper aims to fill this blank and provide a valuable starting point.

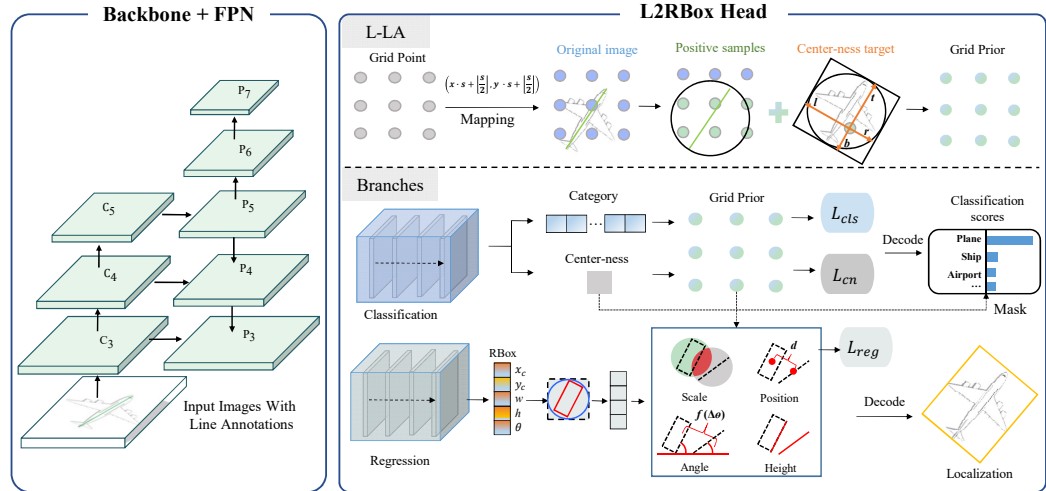

Figure 2: Illustration of proposed L2RBox. The Backbone+FPN extract features from input images and then fed into L2RBox Head. Line annotation-based label assignment (L-LA) of the Head contains training sample selection and center-ness target computation. Branches of the Head contain classification and regression branches. The regression loss is designed as four components.

## 3 PROPOSED METHOD

In this section, we introduce the first implementation of oriented object detection using line as supervised information. First, an overview of our L2RBox is provided in section 3.1. Next, we introduce the line annotation-based label assignment (L-LA) containing training sample selection and centerness target computation in sections 3.2 and 3.3. Finally, we present the proposed loss functions, which guide the optimization of RBox based on line annotations in section 3.4.

### 3.1 OVERVIEW

For simplicity and efficiency, our L2RBox follows the FCOS-based Tian et al. (2019) detection paradigm, which features a one-stage, anchor-free architecture. The overview of the L2RBox is shown in Fig. 2. Our L2RBox initially leverages a backbone and feature pyramid network (FPN) Lin et al. (2017a) to extract multiscale features from original images. Notably, our detector is adaptable to various backbone networks, including ResNet He et al. (2016), Swin Transformer Liu et al. (2021), and ConvNeXt Liu et al. (2022). Unless otherwise specified, all methods in this paper use ResNet50 by default for fair comparison.

Following FPN, we detect objects of different sizes at various feature map levels. Each pyramid level, denoted by $P_i$, corresponds to a feature level. Specifically, $P_3$, $P_4$, and $P_5$ are derived from the backbone CNN's feature maps $C_3$, $C_4$, and $C_5$, followed by a $1 \times 1$ convolutional layer with top-down connections. $P_6$ and $P_7$ are generated by applying a convolutional layer with a stride of 2 on $P_5$ and $P_6$, respectively. Each feature pyramid level $\{P_3, P_4, P_5, P_6, P_7\}$ is then fed directly into the L2RBox Head.

The L2RBox Head consists of two branches. The classification branch predicts the confidence scores, while an additional single-layer convolution predicts the center-ness of the location in parallel. The other branch is responsible for RBox regression, enabling loss computation between the predicted RBox and the ground truth lines. The total loss is calculated as the weighted sum of the classification and regression losses. Furthermore, we propose a line annotation-based label assignment strategy (L-LA) that contains training sample selection and center-ness target computation.

### 3.2 TRAINING SAMPLE SELECTION FOR L2RBOX

Given a feature map $F \in \mathbb{R}^{H \times W \times C}$, where $H$, $W$, and $C$ correspond to the height, width, and channel of the feature map, respectively, the set of grid locations for this feature map is represented

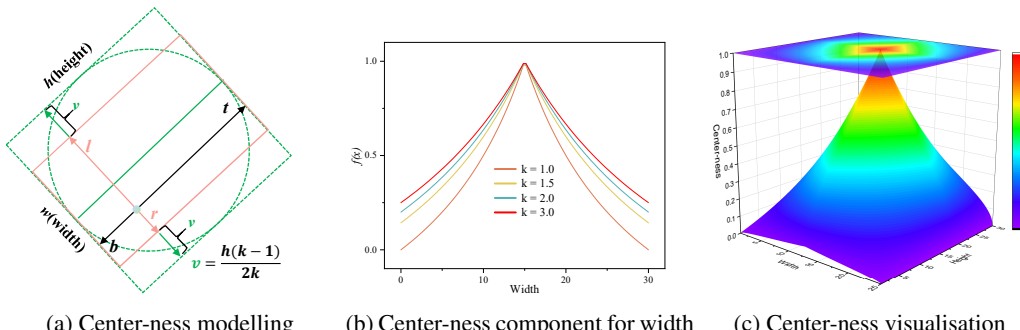

(a) Center-ness modelling  (b) Center-ness component for width  (c) Center-ness visualisation

Figure 3: Illustration of the center-ness for L2RBox. (a) illustrates the center-ness modeling process using line annotations, where green represents the supervised information and orange shows the actual bounding box. (b) compares the trend curves of center-ness component across various scales. (c) visualization of the center-ness values and projection on the $xy$-axis.

as $\mathbb{P} = \{(x_i, y_i)|i = 1, 2, 3, ..., H \times W\}$. The correspondence between any point $(x_i, y_i)$ in the set $\mathbb{P}$ and the original image position $(x_i^a, y_i^a)$ is as follows:

$$(x_i^a, y_i^a) = (\left\lfloor \frac{s}{2} \right\rfloor + x_i \cdot s, \left\lfloor \frac{s}{2} \right\rfloor + y_i \cdot s) \tag{1}$$

where $s$ denotes the stride of feature map $F$. The $j$th ground truth line for an input image is defined as $L_j = (x_0^j, y_0^j, x_1^j, y_1^j, c^j) \in \mathbb{R} \times \{1, 2, ..., C\}$, where $(x_0^j, y_0^j)$ and $(x_1^j, y_1^j)$ are the endpoints of the line, and $c^j$ is the class that the line annotation object belongs. $C$ is the number of classes, which is 15 for DOTA-v1.0 dataset. Different from bounding box-supervised approaches that can accurately define the object boundaries, our method requires expanding the line into a circular region $C_g$ to approximate the boundaries. The centre $(x_c, y_c)$ and radius $R$ of circular region $C_g$ are defined as follows:

$$(x_c, y_c) = (\frac{x_0^j + x_1^j}{2}, \frac{y_0^j + y_1^j}{2}), R = \sqrt{(\frac{x_1^j - x_0^j}{2})^2 + (\frac{y_1^j - y_0^j}{2})^2} \tag{2}$$

To ensure the quality of the training samples, we perform center sampling on the circular region $C_g$ with a sampling radius $R_c = s \times o$, where $s$ denotes the stride and $o$ denotes the sampling ratio. The sampling result is denoted as region $C_c$. We then obtain the set $\mathbb{P}_g$ and $\mathbb{P}_c$, belonging to $C_g$ and $C_c$ in the following way:

$$\mathbb{P}_g = \left\{(x_i, y_i)|i \in \left\{i|(x_i^a - x_c)^2 + (y_i^a - y_c)^2 < R^2\right\}\right\},$$
$$\mathbb{P}_c = \left\{(x_i, y_i)|i \in \left\{i|(x_i^a - x_c)^2 + (y_i^a - y_c)^2 < R_c^2\right\}\right\} \tag{3}$$

We choose not to directly use the center sampling region for selecting training samples because the center sampling radius must be globally adjusted based on the stride of each output layer, which can result in the radius exceeding the circular region. Instead, the final positive samples set $\mathbb{P}^+ = \mathbb{P}_g \cap \mathbb{P}_c$, and the labels of positive samples determined by the corresponding GT line. Although objects with different sizes are assigned to different feature levels, densely arranged objects at the same level can still result in a location being assigned to more than one circular region. In such cases, we select the ground truth line with the shortest length as the target.

### 3.3 CENTER-NESS FOR L2RBOX

The FCOS-based detector uses the predict center-ness as a mask to eliminate low-quality predict bounding boxes produced by locations far away from the center of an object. The optimisation target for center-ness is determined by the distances $(l, r, t, b)$ from $(x_i^a, y_i^a)$ to the four edges of the RBox $B_r(x^*, y^*, w, h, \theta)$. The calculation formula is as follows:

$$l = (x_i^a - x^*)cos\theta + (y_i^a - y^*)sin\theta + \frac{w}{2}, r = (x_i^a - x^*)cos\theta + (y_i^a - y^*)sin\theta - \frac{w}{2},$$
$$t = -(x_i^a - x^*)sin\theta + (y_i^a - y^*)cos\theta + \frac{h}{2}, b = -(x_i^a - x^*)sin\theta + (y_i^a - y^*)cos\theta - \frac{h}{2} \tag{4}$$

where $(x^*, y^*)$ is the center, $w$, $h$, and $\theta$ are the width, height, and angle of the RBox, respectively.

Unlike box annotations that can compute all four distances $(l, r, t, b)$, line annotations are limited to calculating only two: $t$ and $b$. To overcome this limitation, as shown in Fig. 3a, we generate a circle with the midpoint of the line as its center and the line segment as its diameter. The minimum circumscribed rectangle of the circle, aligned with the line, is then used to approximate center-ness. The width of a real box can then be considered as a scaling of the line in different ratios $k$. Our approximate center-ness $cn$ can be expressed as follows:

$$cn = \sqrt{\frac{min(l,r) + v}{max(l,r) + v} \times \frac{min(t,b)}{max(t,b)}}, v = \frac{h(k-1)}{2k} \tag{5}$$

According to Eq. 5, we focus on comparing the center-ness component for width, which shows a consistent trend at various ratios, as illustrated in Fig. 3b. We also visualize the center-ness values in Fig. 3c, with higher values found closer to the center. This confirms that our modeling approach provides a suitable optimization target for center-ness.

### 3.4 LOSS FUNCTIONS

Since the detector structure is based on FCOS, the losses in this part mainly include the regression $L_{reg}$, classification $L_{cls}$, and center-ness $L_{cn}$. The loss function for L2RBox is defined as follows:

$$Loss = \frac{\mu_1}{N_{pos}} \sum_i L_{cls}(c_i^*, c_i) + \frac{\mu_2}{N_{pos}} \sum_i L_{cn}(cn_i^*, cn_i)$$

$$+ \frac{\mu_3}{\sum cn_{pos}} \sum_i \mathbf{1}_{c_i > 0} cn_i L_{reg} \{B_i, L_i\} \tag{6}$$

where $L_{cls}$ is the focal loss Lin et al. (2017b), $L_{cn}$ is cross-entropy loss, and $L_{reg}$ is regression loss for L2RBox. $N_{pos}$ denotes the number of positive samples. $c^*$ and $c$ denote the probability distribution of various classes calculated by Sigmoid function and target category. $B$ and $L$ represent the predict RBox and the GT line, respectively. $cn_i^*$ and $cn_i$ indicate the predict and target center-ness. $\mathbf{1}_{c_i > 0}$ is the indicator function, being 1 if $c_i > 0$ and 0 otherwise.

We present the $L_{reg}$ to compute the regression loss between $L(x_c^*, y_c^*, h^*, \theta^*)$ and the predict RBox $B(x_c, y_c, w, h, \theta)$. As shown in the regression branch in Fig. 2, the proposed $L_{reg}$ comprises four components: scale loss $L_s$, height loss $L_h$, position loss $L_p$ and angle loss $L_a$.

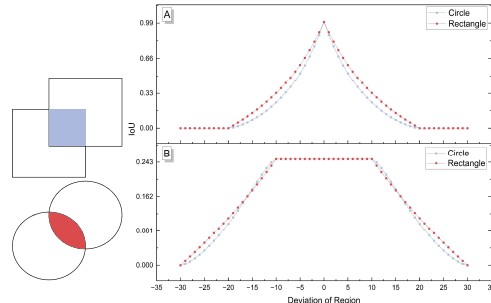

Figure 4: IoU curves for circles and corresponding minimum circumscribed rectangles. A and B are two cases with similar scales and large-scale gaps, respectively.

**Scale loss:** The optimization objective is restricted to the approximate scale of the object by computing the scale loss $L_s$ between the corresponding circles of $B$ and $L$. The formula is as follows:

$$L_s = -\ln \frac{r2c(B_i) \cap l2c(\beta L_i)}{r2c(B_i) \cup l2c(\beta L_i)} \tag{7}$$

where the $r2c(\cdot)$ function converts the RBox into its minimum circumscribed circle, and the $l2c(\cdot)$ function converts the target line into a circle by taking the segment as the diameter. The scale factor $\beta$ influences the scale of the optimization objective through the scaling line, ranging from $[1, \sqrt{2}]$ and reaching its peak of $\sqrt{2}$ when the bounding box is square. To simplify the computation of the loss function, motivated by the core concept of KFIoU Yang et al. (2022), which emphasizes constructing loss functions that maintain trend-level consistency with the objective function, we calculate the IoU loss based on the minimum horizontal circumscribed rectangles of the circles. As shown in Fig. 4, there is trend-level consistency between the intersection over union (IoU) of circles and the IoU of their minimum horizontal circumscribed rectangles, and we also give the proof procedure in A.1.

**Height loss:** Leveraging supervised information to direct the optimization of the long edges of the predict bounding box. The formula is as follows:

$$L_h = l_1 \{h^*, h\} \tag{8}$$

where $l_1$ is the mean absolute error loss.

**Position loss:** When the IoU loss remains constant (case B, Fig. 4), $L_p$ guides optimization using center distance. The formula is as follows:

$$L_p = \sum_{t \in (x,y)} l_1(t^*, t) \tag{9}$$

where $t$ and $t^*$ represent the centers of the line and the bounding box, respectively.

**Angel loss:** $L_a$ takes into account the loss discontinuity caused by the periodicity of angle. The formula is as follows:

$$L_a = l_1(\sin(\theta - \theta^*), 0) \tag{10}$$

The regression loss $L_{reg}$ is calculated as a sum of the $L_s$, $L_h$, $L_p$ and $L_a$ by the following equation:

$$L_{reg} = \alpha L_s + (1 - \alpha)L_h + L_p + L_a \tag{11}$$

where $\alpha$ is an adjustment factor designed to correlate height with scale.

# 4 EXPERIMENT

## 4.1 DATASETS AND IMPLEMENTATION DETAILS

**DOTA-v1.0:** DOTA-v1.0 Xia et al. (2018) is a commonly used dataset in the field of oriented object detection, which contains 2806 large-scale images. The dataset encompasses the following 15 distinct object classes: Bridge (BR), Harbor (HA), Ship (SH), Plane (PL), Helicopter (HC), Small vehicle (SV), Large vehicle (LV), Baseball diamond (BD), Ground track field (GTF), Tennis court (TC), Basketball court (BC), Soccer-ball field (SBF), Roundabout (RA), Swimming pool (SP), and Storage tank (ST). The proportion of the training set, validation set, and testing set are 1/2, 1/6, and 1/3, respectively. For training and testing, we adhere to a standard protocol by cropping images into $1,024 \times 1,024$ patches with a stride of 824. The detection accuracy is obtained by submitting testing results to DOTA's evaluation server.

**DIOR-R:** DIOR-R Cheng et al. (2022) is an aerial image dataset. Different imaging conditions, weather, seasons, and image quality are the major challenges of DIOR-R. Besides, it has high inter-class similarity and intra-class diversity. The dataset comprises 190,288 objects of interest across 20 categories, totaling 23,463 optical images collected from Google Earth. The categories are defined as: Airplane (APL), Airport (APO), Baseball Field (BF), Basketball Court (BC), Bridge (BR), Chimney (CH), Expressway Service Area (ESA), Expressway Toll Station (ETS), Dam (DAM), Golf Field (GF), Ground Track Field (GTF), Harbor (HA), Overpass (OP), Ship (SH), Stadium (STA), Storage Tank (STO), Tennis Court (TC), Train Station (TS), Vehicle (VE) and Windmill (WM).

**Line Annotation:** To accurately reproduce the biases during manual annotation, we apply random translations and rotations to the labels. Translations ranges are set to 10%, 20%, and 40% of the line length, while rotations ranges are limited to 10%, 20%, and 40% of $\pi/2$. The effect of the range will be discussed in Section 4.3.

**Experimental Settings:** All methods are implemented under the open-source PyTorch 1.13.1 Paszke et al. (2019) framework and the rotation detection tool kits: MMRotate 1.0.0 Zhou et al. (2022). For a fair comparison, all models are configured based on ResNet50 He et al. (2016) backbone and trained on Tesla A100-40g GPUs. The models are optimized using the AdamW optimizer Loshchilov & Hutter (2017), with an initial learning rate of 1e-4 and a mini-batch size of 2. $AP_{50}$ is selected as the main metric for comparison with existing methods. $AP_{75}$ and AP are more stringent evaluation metrics, where AP refers to $AP_{50:95}$, as commonly used in the object detection field. "3x" schedule indicates 36 epochs, with 12 epochs being the default. "MS" and "RR" denote multi-scale technique and random rotation augmentation. Random flipping is employed to prevent over-fitting.

**Trade-offs Metric:** Multi-criteria decision-making Zlaugotne et al. (2020) offers a structured approach to select the optimal option while balancing various criteria. Building on this, we develop a trade-off evaluation metric $M$ that accounts for annotation efficiency and detection accuracy:

Table 1: Detection results of each category on the DOTA-v1.0 and the $AP_{50}$ of all categories. 'RC' indicates using rectangles and circles with curve textures as basic patterns. 'SK' indicates using one sketch pattern for each category as basic patterns. 'FCOS' and 'R-CNN' denote the use of pseudo-labels generated by PointOBB to train FCOS and R-CNN detectors respectively.

| Method | PL | BD | BR | GTF | SV | LV | SH | TC | BC | ST | SBF | RA | HA | SP | HC | $AP_{50}$ |
|---|---|---|---|---|---|---|---|---|---|---|---|---|---|---|---|---|
| **RBox-supervised:** | | | | | | | | | | | | | | | | |
| RetinaNet Lin et al. (2017c) | 87.9 | 77.3 | 39.7 | 61.4 | 75.9 | 54.4 | 75.6 | 90.8 | 77.4 | 79.7 | 51.8 | 61.5 | 50.8 | 65.1 | 35.3 | 65.63 |
| RepPoints Yang et al. (2019) | 86.7 | 81.1 | 41.6 | 62.0 | 76.2 | 56.3 | 75.7 | 90.7 | 80.8 | 85.3 | 63.3 | 66.6 | 59.1 | 67.6 | 33.7 | 68.45 |
| KFIoU Yang et al. (2022) | 89.1 | 75.2 | 49.0 | 69.7 | 78.1 | 75.5 | 86.7 | 90.9 | 83.7 | 84.5 | 62.2 | 62.9 | 66.7 | 65.9 | 50.2 | **72.68** |
| FCOS Tian et al. (2019) | 88.4 | 75.6 | 48.0 | 60.1 | 79.8 | 77.8 | 86.6 | 90.1 | 78.2 | 85.0 | 52.8 | 66.3 | 64.5 | 68.3 | 40.3 | 70.78 |
| **HBox-supervised:** | | | | | | | | | | | | | | | | |
| BoxLevelSet-RBox Li et al. (2022b) | 63.5 | 71.3 | 39.3 | 61.1 | 41.9 | 41.0 | 45.8 | 90.9 | 74.1 | 72.1 | 47.6 | 63.0 | 50.0 | 56.4 | 28.6 | 56.44 |
| SAM Kirillov et al. (2023) | 78.6 | 69.2 | 31.4 | 56.7 | 72.2 | 71.4 | 77.0 | 90.5 | 76.2 | 83.7 | 42.5 | 59.5 | 51.1 | 56.2 | 42.9 | 63.94 |
| H2RBox Yang et al. (2023) | 88.5 | 73.5 | 40.8 | 56.9 | 77.5 | 65.4 | 77.8 | 90.9 | 83.2 | 85.3 | 55.3 | 62.9 | 52.4 | 63.6 | 43.3 | **67.82** |
| **Point-supervised:** | | | | | | | | | | | | | | | | |
| P2BNet Chen et al. (2022)+H2RBox | 24.7 | 35.9 | 7.0 | 27.9 | 3.3 | 12.1 | 17.5 | 17.5 | 0.8 | 34.0 | 6.3 | 49.6 | 11.6 | 27.2 | 18.8 | 19.63 |
| P2BNet+H2RBoxv2Yu et al. (2024b) | 11.0 | 44.8 | 14.9 | 15.4 | 36.8 | 16.7 | 27.8 | 12.1 | 1.8 | 31.2 | 3.4 | 50.6 | 12.6 | 36.7 | 12.5 | 21.87 |
| Point2RBox-RC Yu et al. (2024a) | 62.9 | 64.3 | 14.4 | 35.0 | 28.2 | 38.9 | 33.3 | 25.2 | 2.2 | 44.5 | 3.4 | 48.1 | 25.9 | 45.0 | 22.6 | 32.92 |
| Point2RBox-SK Yu et al. (2024a) | 53.3 | 63.9 | 3.7 | 50.9 | 40.0 | 39.2 | 45.7 | 76.7 | 10.5 | 56.1 | 5.4 | 49.5 | 24.2 | 51.2 | 33.8 | **40.27** |
| PointOBB Luo et al. (2024) (FCOS) | 26.1 | 65.7 | 9.1 | 59.4 | 65.8 | 34.9 | 29.8 | 0.5 | 2.3 | 16.7 | 0.6 | 49.0 | 21.8 | 41.0 | 36.7 | 30.08 |
| PointOBB Luo et al. (2024) (R-CNN) | 28.3 | 70.7 | 1.5 | 64.9 | 68.8 | 46.8 | 33.9 | 9.1 | 10.0 | 20.1 | 0.2 | 47.0 | 29.7 | 38.2 | 30.6 | 33.31 |
| **Line-supervised:** | | | | | | | | | | | | | | | | |
| L2RBox (Ours) | 86.1 | 66.2 | 21.2 | 57.5 | 74.5 | 5.5 | 44.7 | 90.7 | 80.3 | 62.6 | 55.1 | 45.3 | 26.5 | 68.1 | 32.2 | 54.48 |
| L2RBox (Ours) (3×, RR) | 86.2 | 69.5 | 18.7 | 58.6 | 73.7 | 6.4 | 45.3 | 90.5 | 79.4 | 64.4 | 56.9 | 40.9 | 26.8 | 70.7 | 32.5 | 54.66 |
| L2RBox (Ours) (MS) | 88.1 | 70.1 | 23.0 | 62.8 | 80.1 | 6.1 | 47.9 | 90.9 | 83.9 | 70.4 | 64.0 | 47.4 | 23.1 | 71.4 | 43.1 | 58.14 |
| L2RBox (Ours) (MS,RR) | 87.5 | 71.3 | 27.2 | 64.5 | 80.3 | 6.1 | 47.2 | 90.9 | 83.7 | 70.2 | 63.7 | 44.0 | 24.7 | 71.7 | 41.0 | **58.26** |

$$M_i = N_a(A_i) * N_t(T_i) \tag{12}$$

where $T$ denotes the annotation time cost for different labeling methods. $A$ denotes the $AP_{50}$ of different training methods. $N_t(\cdot)$ and $N_a(\cdot)$ refer to the normalizing processes for annotation time cost and detection accuracy, respectively. Notably, we adopt the max-min normalization method Jahan & Edwards (2015) and utilize the fully-supervised method KFIou as the reference:

$$N_a(A_i) = (A_i - 0)/(A_s - 0), \quad N_t(T_i) = (T_s - T_i)/(T_s - 0) \tag{13}$$

where $A_s$ denotes the $AP_{50}$ of the supervised method, $T_s$ denotes the annotation time cost of RBox.

## 4.2 MAIN RESULTS

**Results of trade-off metric $M$.** We take RBox-supervised KFIoU Yang et al. (2022) as the benchmark and calculate the trade-off metric $M$ of different weakly-supervised methods, which have the best performance under HBox- point- and line-supervised, respectively (performance see Table 1). As shown in Fig. 5, red pint, green point, and blue point correspond to our L2RBox, point2rbox Yu et al. (2024a), and H2RBox Yang et al. (2023), respectively. The horizontal axis is normalized time efficiency and the vertical axis is normalized accurancy. The colour area represents $m$. Though the HBox-supervised method has the advantage in accuracy and the Point-supervised method has the advantage in time efficiency, our Line-supervised has the best trade-off effectiveness with a highest trade-off metric score of 0.52. Experimental results show that our L2Rbox can effectively trade-off annotation cost and performance in weakly supervised detectors.

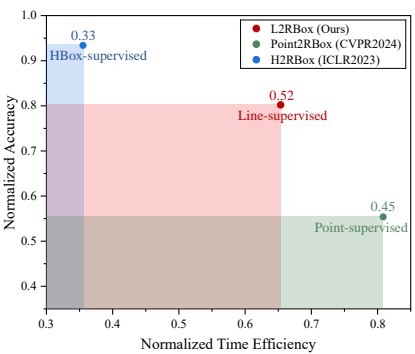

Figure 5: Comparison of trade-off metric for different weakly-supervised methods.

**Results on DOTA-v1.0.** As shown in Table 1, our L2RBox achieves $AP_{50}$ of 54.48%. When multi-scale (MS) technique and random rotation (RR) augmentation are applied, the $AP_{50}$ score reaches up to 58.26%. Compared to Point-supervised approaches, our L2RBox outperforms the best-performing Point2RBox-SK Yu et al. (2024a) by 17.9%. When benchmarked against the FCOS-based PointOBB Luo et al. (2024), the improvement reaches 28.18%. This significant performance advantage results not only from the additional information provided by line supervision over point supervision but also from our network architecture specifically optimized for line supervision. This design enables oriented object detection while fully leveraging the supervision information to enhance detection accuracy. When compared to the more information-rich HBox-supervised method H2RBox Yang et al. (2023), our $AP_{50}$ score is lower by only 9.56%. This difference is much smaller than the improvement our L2RBox achieves over point-supervised methods. Notably, our L2RBox achieves performance comparable to fully RBox-supervised methods in the categories PL, GTF, and TC, and slightly surpasses them in $AP_{50}$ scores for the categories SP, SBF, SV, and BC, demonstrating the potential of line supervision.

**Results on DIOR-R.** To assess the robustness of our L2RBox, we also compared L2RBox against state-of-the-art methods using the DIOR-R dataset, as detailed in A.2.

Table 2: Ablation for main components.

| Dataset | TSS | CN | $L_{reg}$ | AP | $AP_{50}$ | $AP_{75}$ |
|---|---|---|---|---|---|---|
| | ✓ | ✓ | | 0.34 | 1.38 | 0.09 |
| DOTA | ✓ | | ✓ | 12.22 | 29.41 | 8.46 |
| | ✓ | ✓ | ✓ | **21.84** | **54.48** | **17.37** |
| | ✓ | ✓ | | 0.33 | 1.50 | 0.10 |
| DIOR | ✓ | | ✓ | 7.50 | 25.70 | 1.90 |
| | ✓ | ✓ | ✓ | **16.21** | **43.41** | **7.50** |

Table 3: Effect of weight $\mu_3$.

| $\mu_3$ | AP | $AP_{50}$ | $AP_{75}$ |
|---|---|---|---|
| 1.0 | 21.15 | 52.49 | 16.97 |
| 0.7 | 20.62 | 53.75 | **17.68** |
| 0.5 | 21.62 | 54.19 | 17.32 |
| 0.3 | **21.84** | **54.48** | 17.37 |
| 0.1 | 21.51 | 54.43 | 16.76 |

Table 4: Effect of adjustment factor $\alpha$.

| $\alpha$ | AP | $AP_{50}$ | $AP_{75}$ |
|---|---|---|---|
| - | 12.39 | 30.21 | 8.73 |
| 0.3 | 21.22 | 53.90 | 16.60 |
| 0.5 | 21.34 | 53.51 | 17.23 |
| 0.7 | **21.84** | **54.48** | **17.37** |
| 0.9 | 21.47 | 54.17 | 17.21 |

Table 5: Analysis of different scaling ratios.

| $\beta$ | AP | $AP_{50}$ | $AP_{75}$ |
|---|---|---|---|
| 1.00 | 0.10 | 0.14 | 0.00 |
| 1.10 | 21.19 | 46.75 | 14.78 |
| 1.15 | **21.84** | **54.48** | **17.37** |
| 1.20 | 20.53 | 51.65 | 10.52 |
| 1.40 | 12.39 | 30.21 | 8.73 |

## 4.3 ABLATION STUDY

The ablation study is performed on the proposed L2RBox with 12 training epochs.

**The effect of main components.** Table 2 presents the ablation study of the main components. Here, TSS, CN, and $L_{reg}$ represent our proposed training sample selection, center-ness target computation, and regression loss, respectively. TSS is consistently applied to ensure stable network training. "w/o CN" means utilizing only the center-ness component of height, while "w/o $L_{reg}$" means directly optimizing the network with line annotations. The $AP_{50}$ scores are 1.38% and 29.41% on DOTA when adding CN or $L_{reg}$, respectively. When all the components work together, the score achieves 54.43%. These excellent results demonstrate that our proposed method not only enables line-supervised oriented object detection but also significantly enhances detection accuracy. The ablation experiment on the DIOR dataset further supports this conclusion.

**The effect of weight $\mu_3$ of Eq. 6.** We explore the impact of the weight $\mu_3$ of $L_{reg}$ in total loss on the detection performance. Table 3 shows that the overall optimum is achieved when $\mu_3$ is set to 0.3.

**The effect of adjustment factor $\alpha$.** Table 4 examines the effect of the adjustment factor $\alpha$. When $\alpha$ is set to 1 or 0, it indicates that only scale loss $L_s$ or only height loss $L_h$ is used, respectively, both resulting in a detection accuracy of 0. The symbol "-" indicates that the adjustment factor is not applied, meaning that the weights of $L_s$ and $L_h$ are independent of each other. When $\alpha$ is set to 0.7, the $AP_{50}$ score reaches 54.48%, representing a 24.27% improvement compared to when no adjustment factor is applied. This notable performance improvement demonstrates that the proposed adjustment factor effectively links scale and height, jointly guiding the direction of optimization.

**The effect of scale factor $\beta$ of Eq. 7.** We investigate the impact of varying the scale factor $\beta$ on detection performance across its value range. Through a coarse search shown in Table 5, we adopt $\beta = 1.15$. The optimal $AP_{50}$ score shows improvements of 54.34% and 24.27% compared to the scores at the boundaries of the tested range. The results indicate that $\beta$ helps determine the object's scale and, in combination with height information, can return an appropriate bounding box.

**The effect of bias range.** Table 6 displays the effect of different noises. The results show that random translations and rotations with the 20% range slightly improve the performance while the 40% range decreases only 1.92% $AP_{50}$, demonstrating that our method is robust to inaccurate annotations.

Table 6: Ablation studies of bias range. 'T' and 'R' represent the random translation and rotation, respectively. 'T+R' means 'T' and 'R' are used simultaneously.

| Setting | T | R | T + R |
|---|---|---|---|
| Range=0% | 54.43 | 54.43 | 54.43 |
| Range=10% | 54.61 | 54.01 | 54.59 |
| Range=20% | **55.12** | **54.90** | **54.61** |
| Range=40% | 52.77 | 54.46 | 52.51 |

### 4.4 MODEL ANALISIS

**Computational Cost and Speed.** In this study, we examine the computational cost and detection speed of our L2RBox. Given that detection speed is contingent upon the experimental environment, we undertake a comparison of the base architecture FCOS Tian et al. (2019) within our experimental environment, which is equipped with two Tesla A100-40g GPUs. Table 7 illustrates that our method does not result in any additional computational overhead or variation in detection speed. GFLOPs indicate model complexity, with lower values being preferable. "Params" reflects the model size, where fewer parameters are better. Frames per second (FPS) measures inference speed, with higher values being more desirable.

Table 7: Results of computational cost and detection speed.

| Method | GFLOPs | Params(MB) | FPS |
|---|---|---|---|
| FCOS | 206.91 | 31.92 | 27.2 |
| L2RBox(ours) | 206.91 | 31.92 | **27.8** |

**Convergence Analysis.** This section examines the convergence of the proposed model during the training phase. The top row of Fig. 6(a) illustrates the convergence of the total loss and gradient norm curves, which both reach a minimum as training progresses. Furthermore, the loss curves for the classification and regression branches are provided in greater detail in the bottom row of Fig. 6(a). The combined evaluation accuracy curves at different Intersection over Union (IoU) thresholds in Fig. 6(b) demonstrate that the proposed model converges effectively and approximates the global optimum.

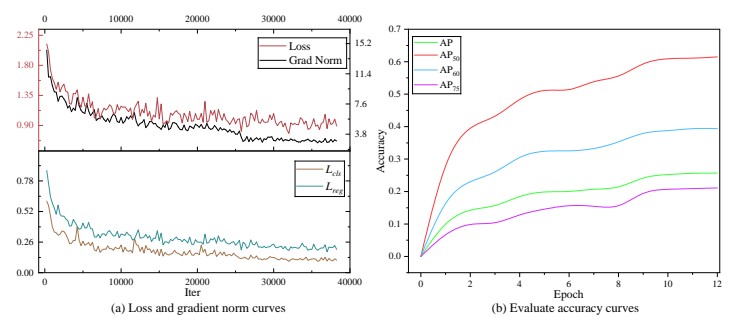

(a) Loss and gradient norm curves     (b) Evaluate accuracy curves

Figure 6: Loss and accuracy curves during training on the DOTA.

## 5 CONCLUSION

This paper introduces a novel line annotation format that balances the annotation cost with the detection accuracy in weakly-supervised oriented object detection methods. We also propose the first line-supervised detector, L2RBox, which includes label assignment and loss functions that support line annotation. The detector employs an anchor-free architecture, enabling end-to-end detection. Extensive experiments demonstrate that our approach significantly outperforms point-supervised methods while requiring only a slight increase in labeling costs.

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

## A Appendix

### A.1 Proof of Trend-level Consistency

This section presents a proof of the trend-level consistency between the Intersection over Union (IoU) of circles and the IoU of their minimum horizontal circumscribed rectangles.

As shown in Fig. 7, suppose there are two circles $\{(x_1, y_1)|x_1^2 + y_1^2 = R_1^2\}$, $\{(x_2, y_2)|(x_2 - d)^2 + y_2^2 = R_2^2\}$, $c_1(x_1, y_1)$ and $c_2(x_2, y_2)$ are the centres of the two circles respectively. $d$ represents the distance between $c_1$ and $c_2$. We establish a Cartesian coordinate system with $c_1$ as the origin and the line between $c_1$ and $c_2$ as the horizontal axis, where $k$ and $p$ are the intersection points of two circles, and $m(x_m, 0)$ is the point where the line segment $kp$ intersect the horizontal axis. The lengths $l_1$ and $l_2$ of the line segments $c_1 m$ and $c_2 m$ are calculated as follows:

$$l_1 = \frac{R_1^2 - R_2^2 + d^2}{2d}, \quad l_2 = \frac{R_2^2 - R_1^2 + d^2}{2d} \tag{14}$$

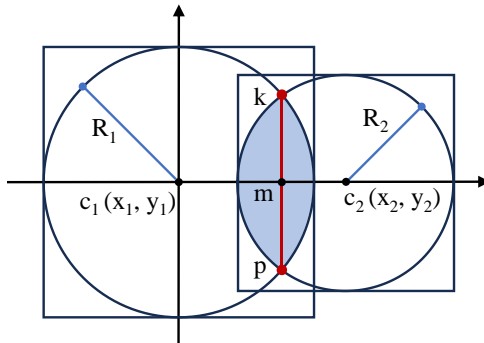

Figure 7: Schematic diagram for calculating Intersection over Union (IoU).

The intersection $I$ of two circles is calculated as follows:

$$
\begin{aligned}
I/2 &= \int_{x_m}^{R_1} \sqrt{R_1^2 - x_1^2}\, dx_1 + \int_{d-R_2}^{x_m} \sqrt{R_2^2 - (x_2 - d)^2}\, dx_2 \\
&= \frac{\pi R_1^2}{4} - \frac{R_1^2 \arcsin \frac{l_1}{R_1} + l_1 \sqrt{R_1^2 - l_1^2}}{2} + \frac{\pi R_2^2}{4} - \frac{R_2^2 \arcsin \frac{l_2}{R_2} + l_2 \sqrt{R_2^2 - l_2^2}}{2}
\end{aligned}
\tag{15}
$$

Then $I$ can be updated with Eq. 14 as:

$$
\begin{aligned}
I &= \frac{\pi R_1^2}{2} - R_1^2 \arcsin \frac{R_1^2 - R_2^2 + d^2}{2dR_1} - \frac{R_1^2 - R_2^2 + d^2}{2d} \sqrt{R_1^2 - \left(\frac{R_1^2 - R_2^2 + d^2}{2d}\right)^2} \\
&+ \frac{\pi R_2^2}{2} - R_2^2 \arcsin \frac{R_2^2 - R_1^2 + d^2}{2dR_2} - \frac{R_2^2 - R_1^2 + d^2}{2d} \sqrt{R_2^2 - \left(\frac{R_2^2 - R_1^2 + d^2}{2d}\right)^2}
\end{aligned}
\tag{16}
$$

When two circles approach each other, the IoU $C_{iou}$ between two circles can be divided into the following cases:

$$
C_{iou} = \begin{cases}
0, & R_1 + R_2 \leq d \\
\dfrac{f(d)}{\pi R_1^2 + \pi R_2^2 - f(d)}, & |R_1 - R_2| < d < R_1 + R_2 \\
\dfrac{\pi \min\{R_1, R_2\}^2}{\pi \max\{R_1, R_2\}^2}, & 0 \leq d \leq |R_1 - R_2|
\end{cases}
\tag{17}
$$

where $f(\cdot)$ denote Eq. 16.

Similarly, the IoU $R_{iou}$ between minimum horizontal circumscribed rectangles of the given circles can be divided into the following cases:

$$
R_{iou} = \begin{cases}
0, & R_1 + R_2 \leq d \\
\dfrac{2\min\{R_1, R_2\}(R_1 + R_2 - d)}{4R_1^2 + 4R_2^2 - 2\min\{R_1, R_2\}(R_1 + R_2 - d)}, & |R_1 - R_2| < d < R_1 + R_2 \\
\dfrac{4\min\{R_1, R_2\}^2}{4\max\{R_1, R_2\}^2}, & 0 \leq d \leq |R_1 - R_2|
\end{cases}
\tag{18}
$$

The monotonicity analysis of $C_{iou}$ and $R_{iou}$ with respect to the variable $d$ is as follows:

Combining the Eq. 17 and Eq. 18, we first discuss the monotonicity when $R_1 = R_2 = R$, where $R$ is an arbitrary constant.

(1) $R_1 + R_2 \leq d$. In this case, $C_{iou} = 0$ and $R_{iou} = 0$.

(2) $|R_1 - R_2| < d < R_1 + R_2$. We calculate the derivative function for $f(d)$ and $R_{iou}$ as follows:

$$
f'(d) = -\sqrt{4R^2 - d^2} < 0, \qquad R_{iou}' = -\frac{4R}{(2R + d)^2} < 0
\tag{19}
$$

Table 8: Detection results of each category on the DIOR-R and the mean $AP_{50}$ of all categories. '1024' and '800' indicate the input images are resized to $1024 \times 1024$ and $800 \times 800$, respectively.

| Method | APL | APO | BF | BC | BR | CH | ESA | ETS | DAM | GF | GTF | HA | OP | SH | STA | STO | TC | TS | VE | WM | $AP_{50}$ |
|---|---|---|---|---|---|---|---|---|---|---|---|---|---|---|---|---|---|---|---|---|---|
| **RBox-supervised:** | | | | | | | | | | | | | | | | | | | | | |
| RetinaNet | 58.9 | 19.8 | 73.1 | 81.3 | 17.0 | 72.6 | 68.0 | 47.3 | 20.7 | 74.0 | 73.9 | 32.5 | 32.4 | 75.1 | 67.2 | 58.9 | 81.0 | 44.5 | 38.3 | 62.6 | 54.96 |
| FCOS | 61.4 | 38.7 | 74.3 | 81.1 | 30.9 | 72.0 | 74.1 | 62.0 | 25.3 | 69.7 | 79.0 | 32.8 | 48.5 | 80.0 | 63.9 | 68.2 | 81.4 | 46.4 | 42.7 | 64.4 | **59.83** |
| **HBox-supervised:** | | | | | | | | | | | | | | | | | | | | | |
| H2RBox | 68.1 | 13.0 | 75.0 | 85.4 | 19.4 | 72.1 | 64.4 | 60.0 | 23.6 | 68.9 | 78.4 | 34.7 | 44.2 | 79.3 | 65.2 | 69.1 | 81.5 | 53.0 | 40.0 | 61.5 | **57.80** |
| H2RBox-v2 | 67.2 | 37.7 | 55.6 | 80.8 | 29.3 | 66.8 | 76.1 | 58.4 | 26.4 | 53.9 | 80.3 | 25.3 | 48.9 | 78.8 | 67.6 | 62.4 | 82.5 | 49.7 | 42.0 | 63.1 | 57.64 |
| **Point-supervised:** | | | | | | | | | | | | | | | | | | | | | |
| P2BNet+H2RBox | 52.7 | 0.1 | 60.6 | 80.0 | 0.1 | 22.6 | 11.5 | 5.2 | 0.7 | 0.2 | 42.8 | 2.8 | 0.2 | 25.1 | 8.6 | 29.1 | 69.8 | 9.6 | 7.4 | 22.6 | 22.59 |
| P2BNet+H2RBox-v2 | 51.6 | 3.0 | 65.2 | 78.3 | 0.1 | 8.1 | 7.6 | 6.3 | 0.8 | 0.3 | 44.9 | 2.3 | 0.1 | 35.9 | 9.3 | 39.2 | 79.0 | 8.8 | 10.3 | 21.3 | 23.61 |
| Point2RBox-SK | 41.9 | 9.1 | 62.9 | 52.8 | 10.8 | 72.2 | 3.0 | 43.9 | 5.5 | 9.7 | 25.1 | 9.1 | 21.0 | 24.0 | 20.4 | 25.1 | 71.7 | 4.5 | 16.1 | 16.3 | **27.30** |
| PointOBB (FCOS) | 58.4 | 17.1 | 70.7 | 77.7 | 0.1 | 70.3 | 64.7 | 4.5 | 7.2 | 0.8 | 74.2 | 9.9 | 9.1 | 69.0 | 38.2 | 49.8 | 46.1 | 16.8 | 32.4 | 29.6 | 37.31 |
| PointOBB (R-CNN) | 58.2 | 15.3 | 70.5 | 78.6 | 0.1 | 72.2 | 69.6 | 1.8 | 3.7 | 0.3 | 77.3 | 16.7 | 4.0 | 79.2 | 39.6 | 51.7 | 44.9 | 16.8 | 33.6 | 27.7 | **38.08** |
| **Line-supervised:** | | | | | | | | | | | | | | | | | | | | | |
| L2RBox (Ours) (800) | 66.7 | 3.2 | 74.2 | 80.7 | 9.2 | 71.9 | 43.6 | 30.3 | 13.3 | 63.6 | 74.9 | 5.7 | 2.0 | 18.7 | 65.1 | 59.9 | 80.4 | 8.2 | 25.0 | 50.7 | 42.40 |
| L2RBox (Ours) (1024) | 73.1 | 4.6 | 74.8 | 80.9 | 9.1 | 71.8 | 40.3 | 34.2 | 11.1 | 64.0 | 77.3 | 4.9 | 2.0 | 19.9 | 69.1 | 66.6 | 80.4 | 5.8 | 28.2 | 50.4 | **43.41** |

Note that as the Eq. 17, the sign of the $C'_{iou}$ is same as the $f'(d)$. This demonstrates that the two functions $C_{iou}$ and $R_{iou}$ exhibit the same monotonicity.

(3) $0 \le d \le |R_1 - R_2|$. According to Eq. 17 and Eq. 18, $C_{iou} = 1$ and $R_{iou} = 1$.

Similarly, we discuss the monotonicity when $R_1 \ne R_2$.

(1) $R_1 + R_2 \le d$. In this case, $C_{iou} = 0$ and $R_{iou} = 0$.

(2) $|R_1 - R_2| < d < R_1 + R_2$. We calculate the derivative function for $f(d)$ and $R_{iou}$ as follows:

$$
\begin{aligned}
f'(d) &= -\frac{\sqrt{-R_1^4 - R_2^4 + 2R_1^2 d^2 + 2R_2^2 d^2 + 2R_1^2 R_2^2 - d^4}}{d} < 0 \\
R'_{iou} &= -\frac{8(R_1^2 + R_2^2)min\{R_1, R_2\}}{[4R_1^2 + 4R_2^2 - 2min\{R_1, R_2\}(R_1 + R_2 - d)]^2} < 0
\end{aligned}
\tag{20}
$$

This demonstrates that $C_{iou}$ and $R_{iou}$ exhibit the same monotonicity.

(3) $0 \le d \le |R_1 - R_2|$. Combine with the Eq. 17 and Eq. 18, $C_{iou} = a$ and $R_{iou} = b$, where $a$ and $b$ are both constants.

In conclusion, regardless of whether the two given circles have the same radius, $C_{iou}$ and $R_{iou}$ exhibit the same monotonicity, indicating a trend-level consistency between the IoU of circles and the IoU of their minimum horizontal circumscribed rectangles.

## A.2 RESULTS ON DIOR-R

To assess the robustness of our L2RBox, we also compared L2RBox against state-of-the-art methods using the DIOR-R dataset, as detailed in Table 8. Our L2RBox achieves $AP_{50}$ score of 43.41%. Compared to the state-of-the-art Point-supervised method (i.e. PointOBB Luo et al. (2024)), our approach uses an end-to-end structure, yet obtains a competitive performance (43.41% vs. 38.08%). When benchmarked against the end-to-end Point2RBox Yu et al. (2024a), our L2RBox demonstrates an improvement of 16.11%. In comparison with fully RBox-supervised methods, L2RBox demonstrates similar performance in the BC, CH, GTF, STO, and TC categories. Notably, in the APL category, our L2RBox outperforms the fully RBox-supervised FCOS Tian et al. (2019) by 11.7%. It is probable that this is a consequence of our utilisation of a minimum circumscribed circle to calculate the loss, which offers greater adaptability for objects with a small aspect ratio.

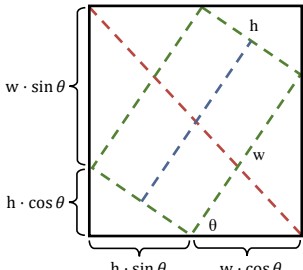

Figure 8: The labeling process for line, HBox and RBox annotation.

### A.3 THEORETICAL COST

We utilize the distance moved while annotating to reflect the overhead in an ideal situation. As shown in Fig. 8, the dotted lines with different colors indicate the trajectory of the different annotation methods. Assuming that the RBox of an object is (x,y,w,h,$\theta$), the distance moved when annotating with RBox can be calculated as:

$$D_r = 2(w + h) \tag{21}$$

When annotating with HBox:

$$D_h = \sqrt{(h \cdot sin\theta + w \cdot cos\theta)^2 + (w \cdot sin\theta + h \cdot cos\theta)^2}$$
$$= \sqrt{w^2 + h^2 + 4sin\theta cos\theta} \tag{22}$$

When annotation with Line:

$$D_l = h \tag{23}$$

$D_r$, $D_h$, and $D_l$ represent the distances moved when using RBox, HBox, and Line annotations, respectively. So the theoretical cost of line annotations is minimal.

Table 9: Ablation studies of sampling ratio

| $o$ | AP | AP$_{50}$ | AP$_{75}$ |
|---|---|---|---|
| 1.0 | 21.45 | 53.49 | 16.93 |
| 1.5 | **21.84** | **54.48** | **17.37** |
| 2.0 | 21.22 | 53.46 | 16.82 |

### A.4 THE EFFECT OF SAMPLING RATIO

We explore the impact of the sampling ratio $o$ introduced in Section 3.2 on detection performance. Table 9 shows that the overall optimum is achieved when $o$ is set to 1.5.

### A.5 QUALITATIVE RESULTS

The qualitative results obtained on the DOTA-v1.0 and DIOR-R are presented in Fig. 9 and Fig. 10. Despite the diverse range of scenes and objects captured in the input images, which encompass a multitude of categories and scales, the proposed L2RBox has demonstrated its capability to accurately predict oriented bounding boxes that are well-aligned with the target. This illustrates that the proposed components can achieve precise oriented object detection based on line annotations.

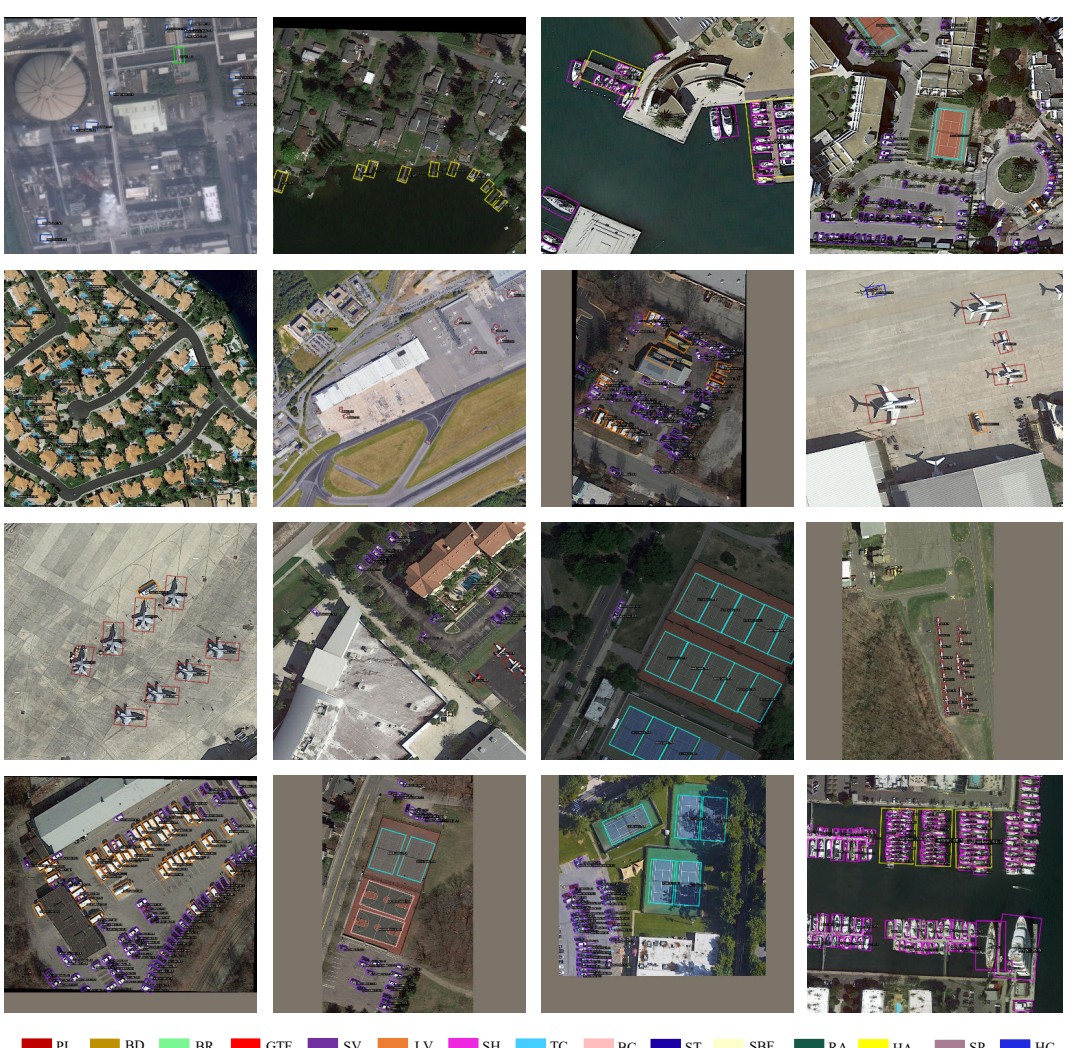

Figure 9: Qualitative Results on DOTA-v1.0.

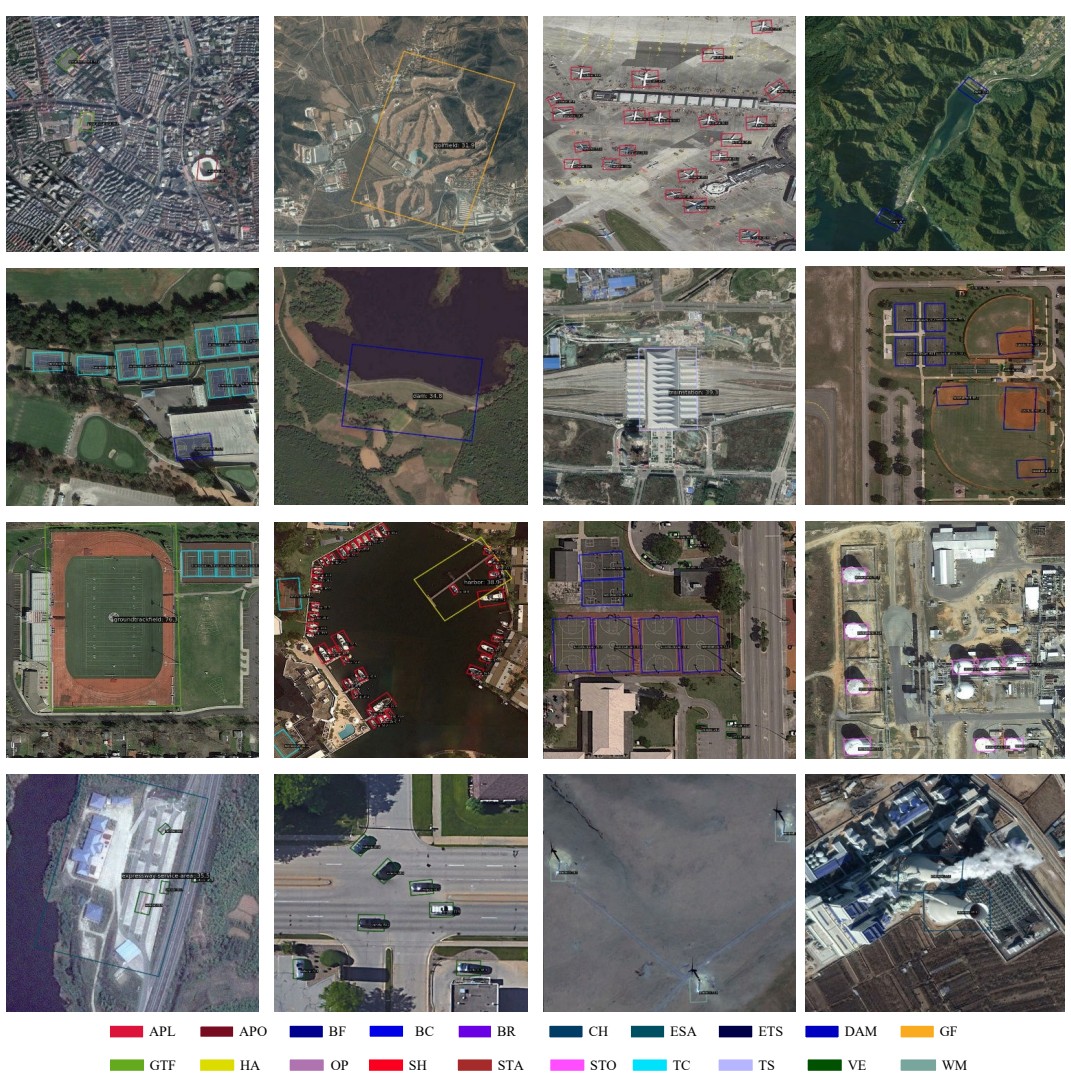

Figure 10: Qualitative Results on DIOR-R.

