# OpenReview forum: "Line2Rbox: Line-supervised Oriented Object Detection"
_ICLR.cc/2025/Conference — ICLR 2025 Conference Withdrawn Submission_

### Official Review · Reviewer_HSm9 · 2024-10-30

**Soundness:** 3
**Presentation:** 3
**Contribution:** 3
**Rating:** 5
**Confidence:** 5

**Summary:**

This paper introduces a novel annotation format for oriented object detection, line annotation, which is intermediate between point-level and box-level annotation. L2RBox is proposed as the first solution for the new proposed task setting, providing a baseline for future research.

**Strengths:**

The research topic is new and the overall writing quality is good. A new solution for weakly supervised oriented object detection based on line annotations is proposed. Experiments on DOTA and DIOR are reported, providing a baseline for future research. If the task setting can be proven useful, it can open up a new field for exploration.

**Weaknesses:**

My major concern is the evidence to prove the meaningness of line-supervision setting is not solid. The main evidence is the claim that the line annotation is faster than HBox (according to https://www.makesense.ai/), but I cannot find such information in the provided link.

**Questions:**

1. Given that both lines and HBoxes are determined by two points, how come lines are much faster than HBoxes?
2. Is the detector trained with line annotations more robust to the annotation inaccuracy than those trained with HBoxes?

---

> ### Author Response · Authors · 2024-11-24
> **Thanks and Response to Reviewer HSm9**
>
> Thanks for your insightful comments. We address your concern below:
>
> **Response to W &Q1&Q2:**
>
> Thank you for your concern about the evidence supporting that line annotation is faster than HBox annotation. We will address this by analyzing four key aspects. In the fourth point, we will simulate annotation bias to evaluate the robustness of L2RBox to inaccurate annotations.
>
> **1)  Statistical analysis**
>
> The average time for annotating 100 instances is 99.15s for point annotations, 178.8s for line annotations, 332.7s for HBox, and 516.2s for RBox. The specific experimental setup is as follows:
>
> - **Dataset:** The DOTA dataset was used for this experiment to ensure consistency.
> - **Tool:** The website [https://www.makesense.ai/](https://www.makesense.ai/) was used for labeling, and the time taken for each method was recorded.
> - **Experts:** A professional annotator with experience in object detection labeling tasks participated.
> - **Annotators:** A group of volunteers with a background in computer vision to perform labeling.
> - **Process:** First, we select some typical images and call for experts to annotate them, establishing an annotation guide. Based on this guide, we train volunteers to perform large-scale labeling. Each annotator labeled a fixed number of instances for each annotation type. Finally, we called for experts and volunteers to double-check each image to filter out low-quality annotations and the average time per 100 instances of annotators was calculated.
>
> **2)  Theoretical cost**
>
> We use the distance moved when annotating to reflect the overhead in an ideal situation.  Assuming that the RBox of an object is (x,y,w,h,$\theta$), then:
>
> Annotate with RBox,
>
> $D_r = 2(w+h)$.
>
> Annotate with HBox,
>
> $D_h =  \sqrt{w^{2}+h^{2}+4sin\theta cos\theta }$.
>
> Annotate with Line,
>
> $D_l = h$.
>
> Where $D_r$, $D_h$, and $D_l$ represent the distances moved when using RBox, HBox, and Line annotations, respectively. $D_r$>$D_h$>$D_l$, so the theoretical overhead of line annotations is minimal.
>
> **3)  Visual distraction**
> - In densely arranged scenes, the HBoxes may contain neighboring objects, visually distracting the labeling process of these objects and thus significantly degrading the annotation efficiency.
>
> - Unlike line annotation, labeling HBoxes requires finding the horizontal circumscribed rectangle of the oriented object.
> The reference information provided by the object during this process is considerably weaker compared to labeling lines within the object.
>
> **4)  Flexible annotation**
>
> The process of line annotations is flexible, allowing for some margin of bias in practice, which can help improve annotation efficiency. To accurately reproduce the biases during manual annotation, we apply random translations and rotations to the labels. Translations ranges are set to 10%, 20%, and 40% of the line length, while rotations ranges are limited to 10%, 20%, and 40% of $\pi$/2. The experimental results are presented below:
>
> **Table A: Ablation studies of the annotation bias. 'Translation+Rotation' means random translation and rotation are used simultaneously.**
> | Setting|Translation|Rotation|Translation+Rotation|
> | --- | :---: | :---: | :---: |
> |Range=0%|54.43|54.43|54.43|
> |Range=10%|54.61|54.01|54.59|
> |Range=20%|**55.12**|**54.90**|**54.61**|
> |Range=40%|52.77|54.46|52.51|
>
> Table A shows that the introduction of appropriate noise proves to be advantageous. The results show that random noise with the 20\% range slightly improves the performance while the 40% range decreases only 1.92\% AP$_{50}$（Translation+Rotation) demonstrating that our method is robust to inaccurate annotations.

---

> > ### Comment · Reviewer_HSm9 · 2024-11-27
> >
> > Thank you for your response. Your explanation is reasonable, particularly regarding the "visual distraction" factor. However, it is concerning that the statement in the rebuttal differs significantly from the original manuscript.
> >
> > **Original text:**
> > According to the labeling website [1], the average time for annotating 100 instances is 99.15s for point annotations, 178.8s for line annotations, 332.7s for HBox, and 516.2s for RBox.
> >
> > The original version implies that the annotation times are based on **third-party statistics**, which makes the data appear highly credible. If, however, these data were actually collected by the authors themselves, then the original statement is misleading. This could give the impression that the authors are concealing this detail and, **through a false claim**, shifting the responsibility of proving the superiority of Line annotations onto an external source.

---

> ### Comment · Reviewer_HSm9 · 2024-11-27
>
> May I ask a follow-up question regarding your experiments on annotation times (Points: 99.15s / 100 instances; RBox: 516.2s / 100 instances):
>
> **If I ask annotators to label the four corner points of 100 tennis courts, should the expected annotation time be 4*99.15s or 516.2s?**
>
> We know that an RBox can be labeled by three points, so I’m having difficulty understanding why annotating three points (3*99.15s / 100 instances) would take much less time than annotating an RBox (516.2s / 100 instances), especially considering that the three points require three classification labels, while the three points of an RBox share a single classification label.

---

> ### Author Response · Authors · 2024-11-28
>
> >... Your explanation is reasonable, particularly regarding the "visual distraction" factor. However, it is concerning that the statement in the rebuttal differs significantly from the original manuscript....
>
>
> - We apologize for the inaccurate expression in the original manuscript. For a fair comparison, we use the annotation website [https://www.makesense.ai/](https://www.makesense.ai/) to evaluate different annotation formats. This has been clarified in the revised manuscript.
>
> - **Fitts's law** [1] indicates that the time required to rapidly move to a target area is a function of the ratio between the distance to the target and the width of the target. The further the distance, the longer the time. Theoretical calculations show that line annotation requires shorter movement distances than HBox or RBox annotation, which explains its reduced time cost.
>
> - Line annotation has practical advantages over HBox annotation. It effectively avoids the **visual distractions** that can occur when using HBox annotation.
>
> - Our L2RBox demonstrates some robustness to inaccurate annotations, increasing the flexibility of line annotations and reducing annotation cost.
>
> [1] The information capacity of the human motor system in controlling the amplitude of movement (1954)

---

> ### Author Response · Authors · 2024-11-28
>
> > May I ask a follow-up question regarding your experiments on annotation times...
>
> While we understand your concern that obtaining precise line annotations can be demanding in terms of annotation costs, please let us clarify a few key aspects:
>
> - Labeling an RBox (oriented bounding box) is not equivalent to labeling 3 or 4 individual points. When annotating an RBox, the annotator needs to carefully adjust the box orientation to fit the object tightly, which requires more time and precision than simply marking point locations. The RBox annotation time of 516.2s per 100 instances captures this additional effort.
>
> - DOTA-v1.0[1] and DOTA-v2.0[2] are well-established and highly-regarded benchmarks for assessing the performance of oriented object detection methods. In these datasets, each instance's location is annotated by a quadrilateral bounding boxes, which can be denoted as "$(x_1, y_1, x_2, y_2, x_3, y_3, x_4, y_4)$" where $(x_i, y_i)$ denotes the positions of the oriented bounding boxes' vertices in the image. The vertices are arranged in a clockwise order.
>
> - Comparing the time to label 3 individual points (3*99.15s) to an RBox (516.2s) is not an apples-to-apples comparison. Point annotations in this context refer to labeling object centers, not box corners. Labeling object centers is much quicker than carefully orienting a bounding box.
>
> - The classification labels are a separate consideration from the localization annotations (points, lines, boxes). The annotation times focus on the geometric labels, not the class labels. Point annotations in this context refer to labeling object centers, not RBox corners. Center points don't define an RBox.  The number of classification labels is independent of the geometric labels (points, lines, RBox). Each object, whether labeled with a point, line, or RBox, gets one class label.
>
> [1] A Large-scale Dataset for Object Detection in Aerial Images (CVPR, 2018).
>
> [2] Object Detection in Aerial Images: A Large-Scale Benchmark and Challenges (TPAMI, 2021)

---

> ### Comment · Reviewer_HSm9 · 2024-12-02
>
> Based on the response, several issues are revealed:
>
> 1. **Inconsistent annotation accuracy.** The authors didn't compare Line/HBox/RBox under the premise of ensuring similar levels of annotation accuracy. The annotators are instructed to "carefully adjust the box to fit the object tightly" for HBoxes and RBoxes, but they are only asked to "simply mark a coarse point location" for Points and Lines.
>
> 2. **Inconsistent tolerance for inaccuracy.** The authors allow the inaccuracy in Line annotations due to the robustness of Line2RBox. However, in the HBox methods they compared (e.g. H2RBox-v2), the accuracy is 71.11\% when 30\% noise is added, meaning that H2RBox-v2 is also robust to inaccuracy. Given this, the authors should allow annotators to "simply mark a coarse HBox" when evaluating the speed of HBox to keep in line with their Line setting.
>
> 3. **Issues with the moving distance theory.** According to the theory presented, the minimal moving distance occurs when annotators are instructed not to zoom in on the image. When they zoom in to improve accuracy, the time required increases. This implies that comparing the annotation times without ensuring comparable accuracy makes no sense.
>
> In summary, the protocol used to evaluate annotation times is unclear and potentially unfair. It is ambiguous even regarding whether RBoxes are annotated with three or four points. These issues undermine the significance of the task setting and make the results less reliable.

---

> > ### Author Response · Authors · 2024-12-03
> >
> > Thank you for your feedback.  In this work, we focus on examining the potential of line annotations for oriented object detection, presenting a practical approach to addressing this challenging task. Our L2RBox achieves a significant performance advantage over point-supervised methods with only a slight increase in annotation cost. The time cost is only introduced to quantify the annotation cost. Experiments with the newly proposed line annotation and other annotation forms were conducted under the same experimental settings. During the annotation process, all annotators are instructed to be as precise as possible: positioning annotations close to the center of the object (for point annotations), fitting the object (for HBox and RBox annotations), or aligning with the central axis of the object (for line annotations). For experimental settings and statistical results, see 'Response to W &Q1&Q2'. We also analyzed the potential factors that make line annotation less time-consuming than box annotation, as well as the robustness of our L2RBox to inaccurate annotations, indicating that our method can reduce annotation requirements and further improve annotation efficiency.
> >
> > >Inconsistent annotation accuracy. & Inconsistent tolerance for inaccuracy.
> >
> > - All labeling formats are conducted in the same experimental settings. In the annotation process, all annotators are required to be as close as possible to the center of the object (point) and fit the object (HBox, RBox) or the central axis of the object (line).
> >
> > - To clarify, we did not respond with "simply mark a **coarse** point location." While HBox annotation requires adjusting the horizontal bounding box to align with the object, line and center point annotations can be directly marked on the object, simplifying the annotation process significantly.
> >
> > - Based on the data from H2RBox-v2[3],  we observed that the performance of H2RBox and H2RBox-v2 degradation begins when noise starts at 10%.
> >
> > - Table A demonstrates that introducing appropriate noise benefits our L2RBox. Specifically, random noise within a 20% range slightly enhances performance.
> >
> > - The robustness of our L2RBox to inaccurate annotations, indicating that our method can reduce annotation requirements and further improve annotation efficiency.
> >
> >
> > >Issues with the moving distance theory.
> >
> > - Theoretical cost is used to demonstrate that line-based labeling is theoretically more efficient than bounding box-based labeling for the same object.
> >
> > - In our analysis, the theoretical cost involves the movement distance of different annotation forms for the same object. The "zoom-in" effect you mentioned should also be considered for the same object to maintain consistency.
> >
> > Furthermore, the annotation form of RBox is discussed in detail in the literature [1][2], and we also emphasized this in the second point of our previous response.
> >
> > [1] A Large-scale Dataset for Object Detection in Aerial Images (CVPR, 2018).
> >
> > [2] Object Detection in Aerial Images: A Large-Scale Benchmark and Challenges (TPAMI, 2021).
> >
> > [3] H2rbox-v2: Incorporating symmetry for boosting horizontal box supervised oriented object detection (Nips,2023).

---

### Official Review · Reviewer_ZvMW · 2024-10-30

**Soundness:** 3
**Presentation:** 2
**Contribution:** 3
**Rating:** 6
**Confidence:** 4

**Summary:**

In the light of line annotation offers a cost-effective approach with orientation data, this paper presents the L2RBOX network, which employs line annotation for oriented object detection. The detector's architecture is based on FCOS and leverages the minimum circumscribed circle and horizontal rectangles derived from the line annotation to supervise the prediction of size, angle, and position.

**Strengths:**

1) Line-supervised oriented object detection is a promising yet challenging task. Exploring how to balance annotation costs and detection performance through line annotations is also an intriguing area of research.

2) The proposed L2RBox demonstrates performance that is comparable to fully RBox-supervised methods in certain scenarios.

**Weaknesses:**

1) The line annotation provides orientation information but lacks some size details compared to the HBox annotation. The key issue is how to derive the missing size information from the line annotation, which only represents a single axis. This paper assumes that the major and minor axes of the object are equal. The proposed solution is to expand the line annotation into circle and square representations, which are not specifically designed to address the missing size information.

2) The proposed speed-accuracy tradeoff is represented by the product of normalized accuracy and time efficiency. The author should review related literature and discuss the validity of this indicator.

3) The network structure is heavily based on FCOS and lacks validation for generalizability across other detection paradigms, making it challenging to assess the effectiveness of the proposed methods.

**Questions:**

1) Have you considered implementing a specific design to predict the minor axis length (i.e. width)? Using statistical priors for aspect ratios across different object classes or estimating the minor axis length based on contextual information could be effective approaches.

2) It would be helpful if the authors could present prediction accuracy for the major and minor axes separately and provide additional analysis.

3) The scaling ratio parameter "k" is introduced to represent the annotated object width for center-ness calculations. However, it appears that "k" does not contribute to performance. Could you clarify the purpose of this hyperparameter?

4) The author could enhance Fig. 5 by providing additional information. For instance, the trade-off balance could be more effectively illustrated by calculating the area under the accuracy-efficiency scatter curve for different classes or subsets, where applicable.

5) Is this Line-to-RBox method adaptable as a plug-and-play approach for other common detectors, such as RetinaNet or Faster-RCNN?

---

> ### Author Response · Authors · 2024-11-24
> **Thanks and Response to Reviewer ZvMW**
>
> Thank you for your valuable comments. We address your concern below:
>
> **Response to W1&Q1:**
> Thank you for your concern regarding the derivation of size information from line annotations. We agree that addressing the missing size information is a critical challenge in line-supervised tasks. Below, we clarify our approach:
>
> - This work is the first to explore the potential of using line annotations for oriented object detection, offering a practical solution to this complex challenge.
> - Predicting missing information is inherently challenging. Our optimization function defines the maximum circular domain of the object (with a diameter equal to $\sqrt{2}$ times the length of the line) based on supervisory information. It then establishes a relationship between the long side and scale using an adjustment factor, ultimately training the model to approximate the ground truth short side.
> - Ablation experiments on the adjustment factor (Section 4.3) reveal that performance degrades significantly when either scale loss or height loss is omitted, indicating the model has some predictive capability for the short edge. However, the adjustment factor's loose constraints make it less sensitive to aspect ratio variations.
> - For future work, we consider adding a separate short-edge prediction branch (self-supervised branch) and setting up supervisory information through view transformations to achieve a more accurate short-edge learning capability.
>
> **Response to W2:**
>
> Reference [1] explores various multi-criteria decision-making methods, a branch of operational research focused on optimizing outcomes in complex scenarios with multiple indicators and conflicting objectives. Based on this, this study designs a trade-off metric to evaluate the overall optimal method considering both detection accuracy and annotation time cost. A standardization process is essential for multi-criteria decision-making, and this paper employs the maximum-minimum standardization approach from Reference [2] to design trade-off indicators. For the specific formula of the trade-off indicator, see section 4.1.
>
> **Response to W3&Q5:**
>
> - Considering that anchor-based methods require heavy detection heads in oriented object detection, we prefer to construct our L2RBox using the anchor-free paradigm.
> - The one-stage anchor-free structure can ensure low computational cost and high inference speed, which is of great significance for practical applications.
> - To the best of our knowledge, this work is the first to explore the potential of using line annotations for oriented object detection, offering a practical solution to this complex challenge while achieving excellent detection performance.
> - In future work, we consider generalizing the line-to-RBox method to other detection paradigms.
>
> **Response to Q2:**
>
> We understand your concern about the predictive ability of our model regarding the major and minor axes.
>
> - In object detection, model performance is typically evaluated using classification scores and the Intersection over Union (IoU) between the predicted bounding box and corresponding ground truth (GT). A high IoU score necessitates the accurate prediction of the major and minor axes, which is a comprehensive measure.
> - Our reply to **'W1 & Q1'** provided an in-depth explanation of our model's capacity to accurately estimate the missing size information.
>
>
> **Response to Q3:**
>
> Thank you for raising this point. "k" indicates the ratio of the long side to the different short sides. Figure 3(b) visualizes the trend of the center-ness component for width when "k" takes on different values. To avoid misunderstanding, the "Ratio" in Figure 3(b) is replaced with "k" in the revised version, and the relevant description is also corrected.
>
> **Response to Q4:**
>
> Thank you for your suggestion.
> - The primary purpose of Figure 5 is to highlight the trade-offs between annotation cost and detection performance across different supervision types. Figure 5 effectively reflects the trade-off between annotation overhead and detection performance of L2RBox.
> - The average time cost calculated in this paper includes all categories.
> - In practice, labeling of different categories is carried out simultaneously.
> - In future work, we consider calculating the accuracy-efficiency scatter curve for different classes.
>
> [1] Multi-Criteria Decision Analysis Methods Comparison (2020 ESWA).
>
> [2] A state-of-the-art survey on the influence of normalization techniques in ranking: Improving the materials selection process in engineering design (2015).

---

> ### Comment · Reviewer_ZvMW · 2024-11-26
>
> I appreciate the authors for addressing all my concerns during the discussion period.  I will increase my rating to 6.

---

> > ### Author Response · Authors · 2024-11-26
> >
> > We sincerely thank you for your thoughtful feedback and for raising your rating. We greatly appreciate your recognition of the novelty and performance of our work. The suggested modifications will be incorporated into the revised manuscript to enhance its clarity and overall quality. Thank you once again for your support!

---

### Official Review · Reviewer_6Ue5 · 2024-10-31

**Soundness:** 3
**Presentation:** 2
**Contribution:** 3
**Rating:** 5
**Confidence:** 5

**Summary:**

This paper innovatively proposes a new task for oriented object detection: Line-supervised Oriented Object Detection. It leverages existing RBox labels to generate precise central axes and designs an end-to-end anchor-free method based on FCOS. L2RBox uses the length information from line annotations and the feature map stride for sample assignment and designs losses to constrain the learning of target regression in terms of scale, height, position, and angle, achieving better performance than existing point-supervised methods in most categories. However, the writing of the paper needs improvement in several areas. For instance, the explanation of the sample assignment process lacks a clear description of how negative samples are assigned. Additionally, some details are unclear, such as the detailed generation method of line annotations and the discussion of their errors.

**Strengths:**

1. Good originality. This paper introduces a new task setting and provides a detailed analysis of the characteristics of various annotations in Oriented Object Detection. It designs a trade-off metric for evaluating existing weakly-supervised annotations (HBox, Point, and Line), thereby demonstrating that the proposed Line annotation can balance accuracy and efficiency.
2. This paper models the Line-supervised oriented object detection problem as a circle-based optimization problem and provides extensive visualizations and mathematical proofs to support this approach.

**Weaknesses:**

1. Issues about the proposed Line-supervised setting.
- The paper lacks a detailed description of the process for generating Line annotations. In PointOBB and Point2RBox, it is mentioned that random sampling within a certain circular range is employed to simulate the error in manual annotations. However, Line2RBox lacks a discussion of this aspect.
2. Limited generalization capability.
- According to the method description of Line2RBox, it lacks consideration for categories with extreme aspect ratios. Based on quantitative and visual results, Line2RBox performs poorly in BR and LV in DOTA-v1.0, and some categories in DIOR-R like APO and BR. Therefore, Line2RBox may be limited to detecting nearly square-shaped objects, overlooking variations in aspect ratios commonly found in remote sensing targets.
3. Unclear writing.
- Additionally, I would like to understand the meaning of taking the union of circles $C_g$ and $C_c$ when selecting training samples, considering that lines 254-256 indicate that a center radius exceeding the circular region is unfavorable.
- In line 248, the specific value and discussion of the sampling ratio seem to be missing.
- It seems that the term corresponding to the L-LA assignment method shown in Figure 2 does not appear in the main text, which may cause confusion

**Questions:**

1. Lines 80-82: Is the annotation speed based on different labeling methods from the website https://www.makesense.ai/ measured by the authors? If so, how was it specifically carried out?
2. In practical manual annotation, can the angle and long edge achieve sufficient accuracy? What is the approximate error margin in manual annotations? The authors are supposed to provide further explanation in relation to "allowing for some margin of error" mentioned in Line 53.
3. How are negative samples selected? In the original FCOS, an ignore region is defined, where samples in the central area are positive and those in the outer region are negative. However, there appears to be no discussion of negative samples in L2RBox.
4. Does the designed scale loss produce similar effects for all categories? I believe targets with different aspect ratios may present varying levels of difficulty in such a loss optimization. For instance, the performance on the LV (large vehicle) category in DOTA-v1.0 is even lower than that achieved by point-supervised methods.
5. Lines 257-260: When facing densely arranged objects, is it appropriate to select the ground truth line with the shortest length as the target? What if a feature point is closer to a longer line in the feature map?

---

> ### Author Response · Authors · 2024-11-24
> **Thanks and Response to Reviewer 6Ue5 (1/2)**
>
> Thanks for your valuable review which helps us improve this work. We address your concerns below:
>
> **Response to W1&Q2:**
>
> Thank you for pointing out the need for a more detailed description of the process for generating line annotations and the lack of discussion regarding simulated annotation errors.
> - We label the object along its central axis, and the process is flexible, allowing some margin of error in the line annotations.
> - To accurately reproduce the biases during manual annotation, we select random translation and rotation to the labels. Translations ranges are set to 10%, 20%, and 40% of the line length, while rotations ranges are limited to 10%, 20%, and 40% of $\pi$/2. The experimental results of our L2RBox on different annotation bias ranges are presented below:
>
> **Table A: Ablation studies of the annotation bias. 'Translation+Rotation' means random translation and rotation are used simultaneously.**
> | Setting|Translation|Rotation|Translation+Rotation|
> | --- | :---: | :---: | :---: |
> |Range=0%|54.43|54.43|54.43|
> |Range=10%|54.61|54.01|54.59|
> |Range=20%|**55.12**|**54.90**|**54.61**|
> |Range=40%|52.77|54.46|52.51|
>
> Table A shows that the introduction of appropriate noise proves to be advantageous. The results show that random noise with the 20\% range slightly improves the performance while the 40% range decreases only 1.92\% AP$_{50}$（Translation+Rotation) demonstrating that our method is robust to inaccurate annotations.
>
> **Response to W2&Q4:**
>
> Thank you for raising the concern regarding the performance of L2RBox on categories with extreme aspect ratios. We acknowledge that L2RBox shows lower performance in certain categories such as BR and LV in DOTA-v1.0 and APO and BR in DIOR-R.
> - This work is the first to explore the potential of using line annotations for oriented object detection, offering a practical solution to this complex challenge.
> - The lower performance in categories with extreme aspect ratios can be attributed to the inherent simplicity of line annotations. Since line annotations do not explicitly encode detailed object shape information, they may struggle to capture fine-grained variations in extremely elongated or compact objects.
> - Ablation experiments on the adjustment factor (Section 4.3) revealed that performance degraded significantly when either scale loss or height loss was omitted, indicating the model has some predictive capability for the short edge. However, the adjustment factor is loosely constrained and thus not very sensitive to aspect ratio differences.
> - For future work, we consider adding a separate short-edge prediction branch (self-supervised branch) and setting up supervisory information through view transformations to achieve a more accurate short-edge learning capability, which in turn enhances the robustness of targets with different aspect ratios.
>
> **Response to W3:**
>
> >...would like to understand the meaning of taking the union of circles...
>
> The radius of center sampling is set according to the step size of the feature map, for some objects, there is a situation where the radius of center sampling is larger than $C_g$, in which case background noise will be introduced if $C_c$ is used.
>
> >In line 248, the specific value and discussion of the sampling ratio seem to be missing.
>
> We appreciate the concern about the sampling ratio. To address this, we conducted additional experiments:
>
> **Table B: Ablation studies of sampling ratio**
>
> |$o$|AP|AP$_{50}$| AP$_{75}$ |
> |---:|:---:|:---:|:---:|
> | 1.0 |21.45|53.49|16.93 |
> | 1.5 |**21.84**|**54.48**|**17.37**|
> | 2.0 |21.22|53.46|16.82|
>
> Table B shows that the overall optimum is achieved when $o$ is set to 1.5.
>
> > ...corresponding to the L-LA assignment method shown in Figure 2...
>
> Thanks for your suggestion. To avoid confusion, we will clarify this in the main text.

---

> ### Author Response · Authors · 2024-11-24
> **Thanks and Response to Reviewer 6Ue5 (2/2)**
>
> **Response to Q1:**
>
> Thank you for your question regarding the annotation speed mentioned in lines 80–82. The specific experimental setup is as follows:
>
> - **Dataset:** The DOTA dataset was used for this experiment to ensure consistency.
> - **Tool:** The website [https://www.makesense.ai/](https://www.makesense.ai/) was used for labeling, and the time taken for each method was recorded.
> - **Experts:** A professional annotator with experience in object detection labeling tasks participated.
> - **Annotators:** A group of volunteers with a background in computer vision to perform labeling.
> - **Process:** First, we select some typical images and call for experts to annotate them, establishing an annotation guide. Based on this guide, we train volunteers to perform large-scale labeling. Each annotator labeled a fixed number of instances for each annotation type. Finally, we called for experts and volunteers to double-check each image to filter out low-quality annotations and the average time per 100 instances of annotators was calculated.
>
> **Response to Q3:**
>
> Thank you for pointing out the need for clarification regarding the selection of negative samples in L2RBox. In L2RBox, all locations that are not designated as positive samples based on the line annotations are treated as negative samples.
>
> **Response to Q5:**
>
> In L2RBox, each feature level corresponds to a specific receptive field size. When pixels at the same feature level overlap multiple target regions (circular region), the smallest region (shortest line) is chosen as the regression target to better align receptive fields with target sizes. This approach ensures that smaller targets are matched to high-resolution feature layers, while larger targets are matched to low-resolution feature layers.

---

> > ### Comment · Reviewer_6Ue5 · 2024-11-28
> >
> > The authors' response has addressed most of my concerns. However, I still have a few questions:
> >
> > 1. If simulated annotation errors are introduced, simply rotating the central axis is inappropriate. Deviations from manual annotations would increase the length of the lines, resulting in additional annotation time.
> >
> > 2. Based on the above discussion and my practical experience with annotation tools, I think the annotation times collected by the authors are somewhat subjective. The annotation speed difference between Hbox and Line may not be nearly twofold (178.8s vs. 332.7s). Additionally, the authors did not specify the categories of the 100 instances. When annotating categories with large aspect ratios, the difference between the annotation times of HBox and Line will further decrease. If the annotation cost becomes comparable to that of Hbox, the significantly lower performance would be unacceptable.

---

> > > ### Author Response · Authors · 2024-12-01
> > >
> > > >If simulated annotation errors are introduced, simply rotating the central axis is inappropriate. Deviations from manual annotations would increase the length of the lines, resulting in additional annotation time.
> > >
> > > - The simulation error we introduced includes simultaneous random translation and rotation of the central axis and is intended to verify the robustness of our L2RBox against inaccurate labeling within a certain range. We have discussed this in detail in our response to W1&Q2.
> > >
> > > - You mentioned that deviations from manual annotations would increase the length. In actual practice, our lines are labeled on the object, and the error in the labeling length is extremely small when there is an object as a reference.
> > >
> > > - HBox annotation is more prone to annotation errors, as it requires careful adjustment of the horizontal bounding box to fit the object. This process demands more time and precision compared to simply labeling a line directly on the object.
> > >
> > > - Our L2RBox supports flexible annotation. For the annotator, looser restrictions can increase the pass rate, reduce the need for rework, and thus improve annotation efficiency.
> > >
> > > >... When annotating categories with large aspect ratios, the difference between the annotation times of HBox and Line will further decrease...
> > >
> > > While we understand your concern that obtaining line annotations can be demanding in terms of annotation costs, please let us clarify a few key aspects:
> > >
> > > The annotation time cost per 100 instances in our statistics is selected based on the statistical proportion of the aspect ratio in the interval (0,6] with a step size of 0.2 in the literature [1]. This includes 15 categories from the DOTA dataset (see Section 4.1 for category details), such as harbors with large aspect ratios and planes with square shapes. We also compare and analyze the annotation cost of HBox and line in the following aspects:
> > >
> > > **1) Theoretical cost**
> > >
> > > We use the distance moved when annotating to reflect the overhead in an ideal situation. Assuming that the RBox of an object is (x,y,w,h,$\theta$), then:
> > >
> > > Annotate with HBox,
> > > $D_h =  \sqrt{w^{2}+h^{2}+4sin\theta cos\theta }$.
> > >
> > > Annotate with Line, $D_l = h$.
> > >
> > > Where $D_h$ and $D_l$ represent the distances moved when using HBox and Line annotations, respectively. $D_h$ > $D_l$, so the theoretical overhead of line annotations is minimal.
> > >
> > > **2) Visual distraction**
> > > - As shown in Fig. 1  in the original manuscript, the HBoxes may contain neighboring objects in densely arranged scenes, visually distracting the labeling process of these objects and thus significantly degrading the annotation efficiency.
> > > - Unlike line annotation, labeling HBoxes requires finding the horizontal circumscribed rectangle of the oriented object. The reference information provided by the object during this process is considerably weaker compared to labeling lines within the object.
> > >
> > > **3) Flexible annotation**
> > >
> > > The process of line annotations is flexible, allowing for some margin of bias in practice. For the annotator, looser restrictions can increase the pass rate, reduce the need for rework, and thus improve annotation efficiency. The supported bias range is discussed in the response to W1&Q2.
> > >
> > >
> > > Our L2RBox is designed to trade off annotation cost with detection accuracy in weakly-supervised methods. With only a slight increase in annotation cost, it achieves a significant performance advantage over point-supervised methods. Conversely, a small reduction in performance can be traded for a larger annotation cost when compared to HBox-supervised methods. This trade-off is highly valuable in practical applications.
> > >
> > >
> > > [1] A Large-scale Dataset for Object Detection in Aerial Images (CVPR, 2018).

---

> ### Comment · Reviewer_6Ue5 · 2024-12-03
>
> Although the author addresses some of my concerns, after further reviewing the comments from other reviewers, I decided to reduce the rating score. My main concern still lies in **the handling of large aspect ratio categories**, and I believe that Line2RBox cannot effectively address them in the short term. The specific reasons are as follows:
>
> 1. Many categories under oriented object detection scenarios have extreme aspect ratios, and Line2RBox shows a significant drop in accuracy for them, like BR and LV in DOTA-v1.0. This makes the proposed method potentially **difficult to generalize to unique datasets with extreme target aspect ratios**, such as HRSC-2016. I believe the current version of the manuscript does not provide an effective solution, as the author mentioned that they will add a separate short-edge branch in the future,
>
> 2. For objects with large aspect ratios, I believe the difference between the annotation times of HBox and Line will further decrease compared to square objects. Considering the significant drop in performance, this makes the results unacceptable.
>
> 3. HBox annotations miss the angle information, and H2RBox and H2RBox-v2 handle this well, achieving accuracy comparable to RBox supervision. Line annotations combine accurate angle and long edge information, while the short edge is missing. However, Line2RBox lacks consideration for the missing information. I believe that if the handling of the short edge were better, Line2RBox might achieve performance close to RBox supervision in the furture.

---

> ### Author Response · Authors · 2024-12-03
>
> Thank you for your feedback. We maintain that even if the current L2RBox may not comprehensively surpass the fully-supervised methods in the short term, as the pioneering exploration of line supervision, we have demonstrated the immense potential of this new direction. Moreover, line supervision incurs lower annotation costs compared to HBox, underscoring its significant research value.
>
> >...difficult to generalize to unique datasets with extreme target aspect ratios, such as HRSC-2016...
>
> 1. We will expand on this discussion of the short edge in the revised manuscript, clarifying the envisioned improvements.
>
> 2. It should be pointed out that objects with extreme aspect ratios account for a small proportion in most object detection datasets, and our method shows significant advantages on objects with medium aspect ratios (such as achieving 58.26% overall on DOTA ). Therefore, L2RBox still has good generalization ability on general datasets.
>
> 3. Based on your comments, we experimented with our L2RBox on the HRSC-2016 dataset which with extreme target aspect ratios, and the results are shown in Table A.
>
> **Table A: AP$_{50}$ on the HRSC-2016 dataset.**
> | Method|AP$_{50}$|
> | --- | :---: |
> |H2Rbox[1]|7.03|
> |L2RBox|53.60|
>
> - As reported in H2RBox-v2 [2], the AP$_{50}$ score of the initial HBox-2-Rbox method, H2RBox [1], on the HRSC dataset is just 7.03%. Our L2RBox outperforms H2RBox by 46.57% on this dataset.
> - Experimental results demonstrate that our L2RBox generalizes more effectively to the HRSC-2016 dataset compared to H2RBox.
>
> >... I believe the difference between the annotation times of HBox and Line will further decrease compared to square objects...
>
>  For objects with large aspect ratios, the time saved by HBox annotation compared to line annotation may be reduced. However, we want to emphasize that the amount of information in line annotations (angle + long edge scale) is significantly higher than point annotations and close to HBox annotations. Therefore, although the accuracy is currently reduced on targets with extreme aspect ratios, overall, line annotations achieve a better balance between annotation cost and detection accuracy. This can be verified by the trade-off metric we proposed (Eq. 12), that is, the comprehensive trade-off score of line supervision is significantly higher than point supervision (0.52 vs. 0.33), and close to the trade-off score of HBox supervision (0.52 vs. 0.45).
>
>  >...achieving accuracy comparable to RBox supervision...
>
> In our manuscript, we compared L2RBox with some fully supervised methods (such as RetinaNet and FCOS under RBox supervision) primarily to provide a performance upper bound as a reference, helping readers comprehensively assess the performance level of our method. However, we did not intend to imply that L2RBox can universally surpass fully supervised methods. As a weakly supervised method, our main goal is to achieve performance as close as possible to fully supervised methods with lower annotation costs.
>
> Moreover, we would like to emphasize that L2RBox approaches or even surpasses some fully supervised methods on multiple categories. For example, in the APL category of DIOR-R, it outperforms FCOS by as much as 11.7% (see Appendix A.2). This indirectly highlights the potential of the brand-new paradigm of line supervision. In the future, by combining more precise short-edge estimation, we hope to further narrow the gap with fully supervised methods.
>
> [1] H2rbox: Horizontal box annotation is all you need for oriented object detection (ICLR2023).
>
> [2] H2rbox-v2: Incorporating symmetry for boosting horizontal box supervised oriented object detection (Nips,2023).

---

### Official Review · Reviewer_Ksfu · 2024-11-01

**Soundness:** 3
**Presentation:** 3
**Contribution:** 3
**Rating:** 8
**Confidence:** 5

**Summary:**

This work aims to introduce a line-supervised oriented object detection method, representing a new form of weak supervision compared to previous point-level and hbox-level formats. To this end, this paper  present a novel line-supervised framework called L2RBOX, which is based on an anchor-free FOCS design. The primary technical contributions include label assignment and loss calculation utilizing line supervision, along with corresponding solutions. Experimental results demonstrate the effectiveness of the proposed method on the DOTA-1.0 and DIOR-R datasets.

**Strengths:**

1. This work proposes a new line supervision format and provides a comprehensive analysis in comparison to previous supervision formats.
2. The introduction of the effective framework L2RBOX includes detailed module designs, with experimental results on DOTA-1.0 and DIOR-R showcasing the superiority of the proposed method.
3. The paper is well-presented overall.

**Weaknesses:**

1. The line supervision format is based on the central axis annotation of objects. While this may simply the problem, a more randomized line representation would enhance its applicability. Given that this is the first work utilizing line supervision, this limitation is understandable.

2. In the related work section, for completeness of article, the discussion on point supervision should include references to the Point-Mask-RBox methodology, which has been discussed in previous point-supervised methods such as PointOBB and Point2Box.

3. Figure 5, which presents a comparison of trade-offs, would benefit from the inclusion of additional point and RBox-supervised methods.

**Questions:**

I lean towards a positive assessment of this work and offer two suggestions:

1. I recommend that the authors further explore the use of random line supervision rather than relying solely on a central line for practical applications in future research.

2. I encourage the authors to release the complete source code to benefit the advancement of this field.

---

> ### Author Response · Authors · 2024-11-24
> **Thanks and Response to Reviewer Ksfu**
>
> Thank you for your valuable comments and constructive feedback. We address each of your points below:
>
> **Response to W1& Q1:**
>
> Thank you for your feedback regarding the line supervision format and its reliance on central axis annotations.
>
> - To simulate a more randomized line representation, we apply random translations and rotations to the labels. Translations ranges are set to 10%, 20%, and 40% of the line length, while rotations ranges are limited to 10%, 20%, and 40% of $\pi$/2.  For details, please refer to **'Response to Reviewer 8DPM - Response to W4'**.
>
> - Our L2RBox serves as an initial exploration of line supervision for oriented object detection, utilizing central line annotations for simplicity. As suggested, future work will focus on increasing the flexibility of line annotations to further reduce labeling costs and enhance the model's adaptability to diverse object shapes. This effort may involve Dataset relabeling broadening the applicability of line-supervised detection in practical scenarios.
>
> **Response to W2:**
>
> Thanks for your valuable suggestion, we will add a discussion of the Point-Mask-RBox methodology.
>
> **Response to W3:**
>
> Thanks for your suggestion.
>
> - The primary purpose of Figure 5 is to highlight the trade-offs between annotation effort and detection performance across different supervision types. For methods using the same annotation format, the annotation time cost remains consistent. Therefore, we focus on the best accuracy achieved under each annotation type, as reported in Table 1.
> - To provide a fair comparison, RBox-supervised methods KFIoU is used as the baseline for standardization.
>
> **Response to Q2:**
>
> Thanks for your suggestion, the complete source code will be released soon.

---

### Official Review · Reviewer_8DPM · 2024-11-03

**Soundness:** 2
**Presentation:** 2
**Contribution:** 2
**Rating:** 6
**Confidence:** 4

**Summary:**

This paper addresses the challenge of balancing annotation cost and detection accuracy in weakly-supervised oriented object detection. The authors propose a novel method called L2RBox, which is the first line-supervised detector for oriented object detection. The method utilizes line annotations as a form of supervision, which is an intermediate level between point-level and plane-level annotations, aiming to reduce the annotation burden while maintaining high detection performance.

**Strengths:**

1. L2RBox introduces a new line annotation format for oriented object detection, which is a unique approach that sits between point and box annotations, offering a potential middle ground in terms of cost and accuracy.
2. The method presents an end-to-end anchor-free detector that uses line labels for label assignment and loss calculation, which is innovative in the context of weakly-supervised object detection.
3. The proposed regression loss is composed of four components (scale loss, height loss, position loss, and angle loss), which support line annotations as an optimization target, a novel approach in the field.
Advantages:
4. The method achieves comparable or even superior performance to fully-supervised detectors in certain categories, demonstrating its effectiveness.

**Weaknesses:**

1. The method involves complex label assignment and loss calculation mechanisms, which might be more challenging to implement compared to simpler point-supervised methods.
2.  While the method shows promising results on DOTA-v1.0 and DIOR-R datasets, its generalization capability to other datasets with different characteristics is not fully explored in the paper.
3. Although the paper mentions that L2RBox does not increase computational cost significantly, the actual resource requirements for training and inference in real-world applications could be a concern for some users.
4. Obtaining precise line annotations can be exceedingly demanding in terms of annotation costs, and in some scenarios, it may even surpass the difficulty of acquiring accurate bounding box annotations.

**Questions:**

see the weaknesses.

---

> ### Author Response · Authors · 2024-11-24
> **Thanks and Response to Reviewer 8DPM (1/2)**
>
> Thank you for taking the time to provide us with a thorough review. We address the specific weakness and questions next:
>
> **Response to W1:**
>
> **1) On Complexity of Implementation:**
>
> Our L2RBox achieves line-supervised oriented object detection by utilizing meticulously designed label assignment strategies and loss functions. As noted in lines 432–440 of the original manuscript, L2RBox significantly surpasses point-supervised methods in performance with only a slight increase in annotation cost, while performing on par with HBox-supervised approaches. Moreover, L2RBox provides multiple advantages over point-supervised methods:
>
> - **High accuracy.**  Our L2RBox outperforms the best-performing Point-supervised method Point2RBox-SK by 17.99%  AP$_{50}$ score (58.26% vs. 40.27% ).
>
> - **End-to-end structure.** Compared to PointOBB, we do not need the intermediate process of pseudo-label generation.
> - **No synthetic patterns are required.** Compared to Point2RBox, our method does not require the generation of synthetic patterns as additional input.
> - **High training efficiency.** Our L2RBox demonstrates high training efficiency, requiring approximately **2 hours** (12 epochs) on two A100-40 GPUs, compared to over **10 hours** (12 epochs) for PointOBB. Additionally, on two 2080ti GPUs, L2RBox completes training in about **4 hours** (12 epochs).
> - **Low computational cost and high speed.** For a more detailed discussion, please refer to Section 4.4.
>
> **2)  Practical Implementation Considerations:**
>
> We have designed the label assignment and loss calculation components to be modular (L2RBox_Head）and (L2RBox_Loss). The corresponding code will be released soon.
>
> **Response to W2:**
> - DOTA-v1.0 and DIOR-R are well-established and highly-regarded benchmarks for assessing the performance of oriented object detection methods. We follow PointOBB (cvpr2024), H2RBox (ICLR2023), etc., and conduct experiments on these two datasets.
> - DOTA-v1.0 contains high-resolution aerial images with densely distributed objects, significant scale variations, and - complex backgrounds, whereas DIOR-R offers a larger, more diverse set of images and categories across various remote sensing platforms. Our method's consistent performance across these datasets highlights its generalization capabilities in diverse scenarios.
> - In future work, we consider evaluating our method on other datasets to assess its generalization capability further.
>
> **Response to W3:**
>
> One of the notable advantages of L2RBox is its computational efficiency. According to our experimental setup, each GPU requires only 5.12 GB of memory making the method feasible for users with limited computational resources. The corresponding training time refers to “**Response to W1**”.

---

> ### Author Response · Authors · 2024-11-24
> **Thanks and Response to Reviewer 8DPM (2/2)**
>
> **Response to W4:**
>
> While we understand your concern that obtaining precise line annotations can be demanding in terms of annotation costs, we respectfully argue that line annotations are less costly than bounding boxes and that our method is robust enough to accommodate some degree of annotation bias. First, we illustrate the flexibility of line annotation (1), followed by a comparative analysis of annotation costs from three perspectives (2,3,4), as follows:
>
> **1)  Bias range of line annotation**
>
> The process of line annotations is flexible, allowing for some margin of bias in practice. To accurately reproduce the biases during manual annotation, we apply random translations and rotations to the labels. Translations ranges are set to 10%, 20%, and 40% of the line length, while rotations ranges are limited to 10%, 20%, and 40% of $\pi$/2. The experimental results of our L2RBox on different annotation bias ranges are presented below:
>
> **Table A: Ablation studies of the annotation bias. 'Translation+Rotation' means random translation and rotation are used simultaneously.**
> | Setting|Translation|Rotation|Translation+Rotation|
> | --- | :---: | :---: | :---: |
> |Range=0%|54.43|54.43|54.43|
> |Range=10%|54.61|54.01|54.59|
> |Range=20%|**55.12**|**54.90**|**54.61**|
> |Range=40%|52.77|54.46|52.51|
>
> Table A shows that the introduction of appropriate noise proves to be advantageous. The results show that random noise with the 20\% range slightly improves the performance while the 40% range decreases only 1.92\% AP$_{50}$（Translation+Rotation) demonstrating that our method is robust to inaccurate annotations.
>
> **2)  Statistical analysis**
>
> As noted in Lines 81-82 of the original manuscript, the average time for annotating 100 instances is 99.15s for point annotations, 178.8s for line annotations, 332.7s for HBox, and 516.2s for RBox. The specific experimental setup refers to **'Response to Reviewer 6Ue5 - Response to Q1'**.
>
> **3)  Visual distraction**
>
> - As shown in Fig. 1, in densely arranged scenes, the HBoxes may contain neighboring objects, visually distracting the labeling process of these objects and thus significantly degrading the annotation efficiency.
>
> - Unlike line annotation, labeling HBoxes requires finding the horizontal circumscribed rectangle of the oriented object. The reference information provided by the object during this process is considerably weaker compared to labeling lines within the object.
>
> **4)  Theoretical cost**
>
> We use the distance moved when annotating to reflect the overhead in an ideal situation. Assuming that the RBox of an object is (x,y,w,h,$\theta$), then:
>
> Annotate with RBox,
>
> $D_r = 2(w+h)$.
>
> Annotate with HBox,
>
> $D_h =  \sqrt{w^{2}+h^{2}+4sin\theta cos\theta }$.
>
> Annotate with Line,
>
> $D_l = h$.
>
> Where $D_r$, $D_h$, and $D_l$ represent the distances moved when using RBox, HBox, and Line annotations, respectively.
> $D_r$ > $D_h$ > $D_l$, so the theoretical overhead of line annotations is minimal.

---

### Author Response · Authors · 2024-11-24
**To all reviewers**

We sincerely thank all the reviewers for their valuable feedback, which has significantly contributed to improving our work. We are delighted that the reviewers acknowledge the novelty and core contributions of our L2RBox (**acknowledged by all reviewers**), the excellent performance has been recognized (noted by reviewers **8DPM**, **Ksfu**, **ZvMW**), the overall writing quality has been appreciated (recognized by Reviewers **Ksfu**, **HSm9**), the extensive visualizations and mathematical proofs have been affirmed (highlighted by reviewer **6Ue5**).  We greatly appreciate your recognition and will try to address each reviewer's questions thoroughly in our rebuttal. Thank you once again for your support and the time and effort dedicated to reviewing our paper.

---

### Note · Authors · 2025-01-30

I have read and agree with the venue's withdrawal policy on behalf of myself and my co-authors.